# A protein-proximity screen reveals Ebola virus co-opts the mRNA decapping complex through the scaffold protein EDC4

Callie J. Donahue[1,9], Aditi S. Kesari[2,9], Naveen Thakur [3], Ling Wang[2], Sarah Hulsey Stubbs[1], Caroline G. Williams[4], Sandhya Bharti Sharma[2], Cara D. Kirby[5], Daisy W. Leung [5,6], Uma K. Aryal [7,8], Christopher F. Basler [5], Douglas J. LaCount [2] ✉ & Robert A. Davey[1] ✉

The interaction of host and Ebola virus (EBOV) proteins is required for establishing infection. In this study, we use proximity-dependent biotinylation to identify cellular proteins that bind to EBOV proteins encoded by six of the seven viral genes. Hits are computationally mapped onto a human protein-protein interactome and annotated with viral proteins, confirming known EBOV-host protein interactions and revealing previously undescribed interactions and processes. This approach efficiently arranges proteins into functional complexes associated with single viral proteins. Focused characterization of interactions between EBOV VP35 and the mRNA decapping complex shows that VP35 binds the scaffold protein EDC4 through the C-terminal subdomain, with both proteins colocalizing in EBOV-infected cells. siRNA depletion of EDC4, DCP2, and EDC3 reduces virus replication by inhibiting early viral RNA synthesis. Overall, the analytical approach efficiently identifies EBOV protein interactions with cellular protein complexes, providing a deeper understanding of replication mechanisms for therapeutic intervention.

Ebola virus (EBOV) causes severe hemorrhagic fever and imposes a heavy public health burden with increasing outbreak frequency. Within the last decade, the two largest EBOV outbreaks have occurred in Western Africa and the Democratic Republic of the Congo, both marked by high fatality rates[1]. Several promising antibody-based medical countermeasures targeting the virus glycoprotein have recently been developed and approved for use in disease treatment. However, these therapeutics must be

administered early in infection to be effective[2]. A better understanding of virus replication mechanisms provides additional targets for drug development.

EBOV is a negative-sense RNA virus with a genome that contains seven genes bordered by 3' and 5' leader and trailer sequences. RNA synthesis is driven by the viral polymerase L in conjunction with VP35, which together make up the RNA-dependent RNA polymerase (RdRp) complex, and cofactors NP and VP30 which make up the viral

---

[1]Department of Virology, Immunology and Microbiology, Chobanian & Avedisian School of Medicine and National Emerging Infectious Diseases Laboratories (NEIDL), Boston University, Boston, MA 02118, USA. [2]Borch Department of Medicinal Chemistry and Molecular Pharmacology, Purdue University, West Lafayette, IN 47906, USA. [3]Department of Microbiology, Icahn School of Medicine at Mount Sinai, New York, NY 10029, USA. [4]Center for Microbial Pathogenesis, Institute for Biomedical Sciences, Georgia State University, Atlanta, GA 30303, USA. [5]Department of Pathology and Immunology, Washington University School of Medicine, St. Louis, MO 63110, USA. [6]Department of Medicine, Washington University School of Medicine, St. Louis, MO 63110, USA. [7]Proteomics Facility, Bindley Bioscience Center, Purdue University, West Lafayette, IN 47907, USA. [8]Department of Comparative Pathobiology, Purdue University, West Lafayette, IN 47907, USA. [9]These authors contributed equally: Callie J. Donahue, Aditi S. Kesari. ✉e-mail: dlacount@purdue.edu; radavey@bu.edu

ribonucleoprotein. Viral transcription produces a gradient of mRNAs, with abundant transcripts from genes at the 3' end and a gradual decrease for genes closer to the 5' end[3]. Genomic replication initiates at the 3' leader sequence and initially synthesizes a positive-sense antigenomic RNA intermediate. Concurrent production of the negative-sense genomic strand initiates at the complement of the trailer region[3]. Both genomic replication and transcription occur in viral inclusion bodies formed by the accumulation of NP[4] that additionally contain VP35, VP30, L, and RNA[5]. EBOV virions consist of seven proteins, including the glycoprotein (GP), matrix protein VP40, the nucleocapsid proteins VP24, VP35 and NP, which coats the RNA genome, and the additional transcription and replication proteins VP30 and L[6]. Viral transcription and genomic replication initiate following virus entry and uncoating.

EBOV replication depends on the multifunctionality of its own proteins as well as virus-host protein interactions to successfully replicate and evade host antiviral responses. Through interactions with NP, VP35 influences the assembly of viral inclusion bodies[4,7]. VP35 caps the blunt end of viral dsRNA, preventing detection by RIG-I sensing and concurrent IFN-β production[8]. In addition to masking viral RNA detection, VP35 abrogates host cell antiviral responses by interacting with host proteins such as the kinases IKKε and TBK-1, preventing the synthesis of Type I interferons[9]. Several groups have sought to expand the set of EBOV-host protein-protein interactions (PPIs) through systematic interaction screening[10–16]. Each screen has provided large volumes of information regarding individual host protein-virus protein interactions, but host proteins are usually found in functional complexes, not in isolation[17].

In this work, we show that integrating proximity-based biotinylation screening with computational analysis of human PPI networks facilitates the identification of host protein complexes engaged by EBOV proteins. To better characterize host protein complexes engaged by viral proteins, we tag each EBOV structural protein (except the transmembrane glycoprotein) with a promiscuous biotin ligase (BioID2)[18]. This approach enables covalent labeling of proteins in direct contact with or in close proximity to EBOV proteins via proximity-dependent biotinylation. Using this approach, we identify the mRNA decapping complex as an important VP35-associated host machinery, suggesting a role in viral replication. These findings underscore the utility of this method in uncovering key host pathways co-opted by EBOV and provide a framework for identifying protein complexes that could serve as potential antiviral targets.

## Results

### Proximity-dependent biotinylation screen identifies 441 interactions between host cell and EBOV proteins

N- and C-terminal BioID2-tagged constructs (NT- and CT-, respectively) were generated for NP, VP35, VP40, VP30, VP24, and L, with GFP-BioID2 and parental cells included as controls (Fig. 1a). Constructs were expressed in tetracycline-inducible Flp-In T-Rex 293 cells to allow for consistent, controlled expression. However, expression levels varied significantly between constructs, likely due to differences in protein stability (Fig. 1b, complete blots are provided in Supplementary Fig. 1). As observed in prior studies[5,10], the RdRp L protein had notably low expression, with the N-terminal BioID2-tagged version just above the detection threshold and the C-terminal fusion being undetectable. Similarly, N-terminally tagged VP30 and C-terminally tagged VP35 exhibited lower expression levels than their opposite-tagged counterparts. To examine the functionality of the BioID2 fusions, a transcription- and replication-competent virus-like particle (trVLP) minigenome assay was performed with tagged NP, VP30, and VP35 as genome replication support plasmids. N- and C-terminal tagged NP, C-terminal tagged VP30, and N-terminal tagged VP35 supported reporter gene expression at levels significantly above the negative

control (25–100% activity compared to untagged constructs, $p \leq 0.01$, one-way ANOVA, Fig. 1c) while N-tagged VP30 and C-tagged VP35 gave the lowest reporter activity in the trVLP assay (<20% of wildtype activity).

We next assessed whether BioID2 tagging maintained the ability of virus proteins to assemble inclusion bodies, a prominent structure in infected cells where virus proteins, RNA and host proteins coalesce and infectious virus particles are formed. HEK293T cells were transfected with N- and C-tagged constructs for NP, VP35, VP40, VP30, and VP24, and challenged with EBOV. Samples were fixed 20 hours post infection and stained for BioID2 and EBOV VP30 as a marker of infected cells and inclusions. By confocal microscopy, all proteins showed cytoplasmic localization. For N- and C-terminal tagged VP24, VP30 and VP35, cytoplasmic colocalization with virus-expressed VP30 was apparent, demonstrating that tagging did not interfere with normal localization of these proteins (Fig. 1d, with individual color channels shown in Supplementary Fig. 2a). For tagged NP, colocalized inclusions were less frequent, but were evident for the N-tagged construct in cells expressing lower levels of the construct. Furthermore, both constructs formed multiple inclusion body-like structures similar to those reported for native NP expression patterns[19]. Since both NT- and CT-tagged NP retained activity in the trVLP assay (Fig. 1c), it was concluded that both tagged constructs were functional and were retained in later analyses. For VP40, which is not part of inclusion bodies or required for trVLP assay activity in p0 cells, BioID2 staining at the cell plasma membrane was observed for both VP40 constructs (Fig. 1e) with pronounced filamentous structures for the N-terminal tagged construct, suggesting incorporation into budding virions like the wild type protein and consistent with previous reports that VP40 tolerates fusions to its N-terminus[20]. Overall, constructs that gave localization similar to native protein in infected cells and/or gave trVLP assay activity >25% of normal were N- and C-tagged NP, N-tagged VP35 and VP40, and C-tagged VP30 constructs and were included in further analysis.

Biotinylation of cellular proteins in cells expressing single BioID2-tagged viral proteins was activated by incubating cells with excess biotin for 24 hours. Cells were then lysed, and biotinylated proteins were purified on streptavidin beads. Five percent of the purified protein was subjected to western blotting to confirm successful biotinylation and protein capture (Fig. 1f). The pattern and level of biotinylation was consistent between replicates (Supplementary Fig. 2b) but varied between constructs. In general, the amounts of biotinylated host proteins correlated with the expression levels of the BioID2 fusion protein. Despite similar levels of expression, the N-terminal fusion to NP labeled several bands more intensely than the C-terminal construct. These bands did not correspond to the sizes of the most abundant proteins in the mass spectrometry analysis, suggesting they may be breakdown products of BioID2-NP.

To identify biotinylated host proteins, we analyzed the samples by mass spectrometry. Spectral counts from the mass spectrometry analysis for each protein were then analyzed using SAINTExpress to eliminate nonspecifically labeled proteins and to develop a set of high-confidence interactors. The SAINT algorithm normalizes spectral counts to protein length and provides a confidence score based on hit and total protein abundance for each sample compared to controls[21]. Two rankings were made using 1) GFP-BioID2 and unmodified parental Flp-In T-Rex 293 cells as background and 2) unique interactions for each viral protein using all other viral proteins plus GFP and parental cells as background (Supplementary Data 1). Because unique virus-host protein interactions were identified in both analyses, each hit list was combined for further study, yielding a final dataset of 441 interactions (Supplementary Fig. 2c). Previously characterized EBOV-host PPIs were identified, including interactions of NP with protein phosphatase 2 regulatory subunits B alpha, B' gamma, and B' delta (PPP2R2A, PPP2R5C, PPP2R5D)[22]; VP24 with karyopherin proteins (KPNA1, 2, 5 and 6)[23]; VP30 with RBBP6[10] and SRPK2, a kinase that

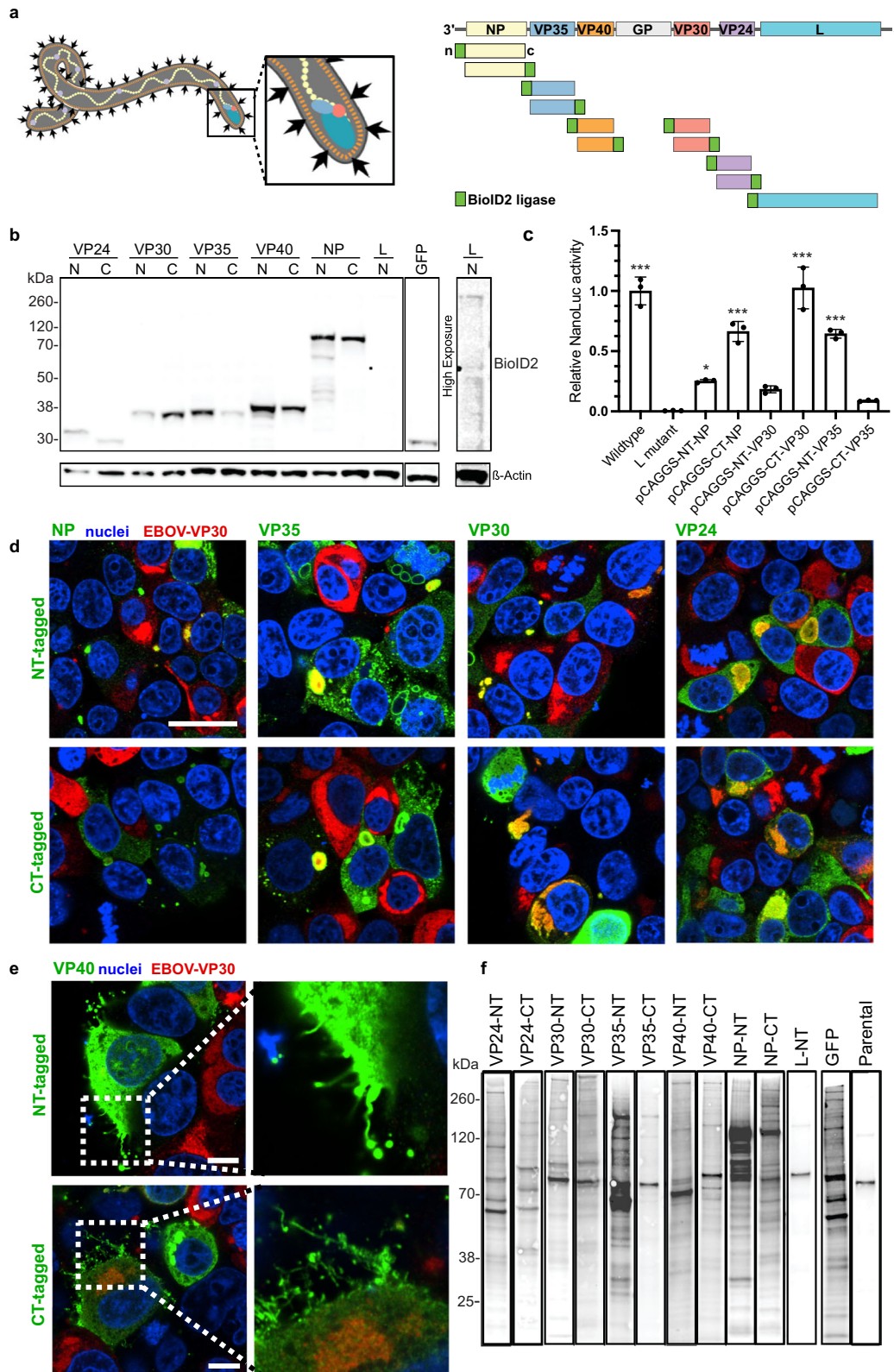

phosphorylates serine residues in the N terminal domain[24], and VP35 with dynein light chain LC8-type 1 and 2 (DYNLL1 and DYNLL2)[25]. Comparisons to other published large-scale studies revealed 35 interactions found in at least one co-affinity purification plus mass spectrometry (co-AP/MS) experiment and 55 from VP40-BioID[15] and L-VP35 split-TurboID[16] screens. Seven hits were implicated in EBOV replication by CRISPR or siRNA screens (Supplementary Data 2)[26,27] and five

proteins were found in EBOV virions by mass spectrometry (Supplementary Data 3)[28]. In total, 81 different interactions were found in at least one other published data set (Supplementary Data 4)[10,14,16,22,24,29,30].

Including both N- and C- terminal BioID2 tags allowed the capture of interactions dependent on intact termini. Of the 441 protein interactions identified, 64 were seen with both N- and C-terminal tagged constructs, and therefore independent of the viral protein N- and

**Fig. 1 | A protein proximity screen for Ebola virus proteins identifies an expanded viral interactome. a** Left: Schematic of structural protein arrangement in the virion. Right: EBOV genome denoting virally encoded genes and screening constructs showing BioID2 ligase (green blocks) at the N or C-termini of each protein. **b** Expression of viral protein-BioID2 fusion constructs and GFP-BioID2 non-specific control in stable cell lines. Blots were probed with biotin ligase-specific antibody, and loading levels were controlled by detection of β-actin. Blots were performed twice with similar results. **c** Tagged proteins were evaluated for trVLP activity by replacing constructs encoding WT proteins with the tagged protein at the first step to generate trVLPs, and the resulting NanoLuc activity was measured. NanoLuc activity was compared to an RdRp L active site mutant using one-way ANOVA with multiple comparisons correction. Data are presented as mean values from three biological replicates +/- standard deviation (SD). * $p = 0.01$, *** $p < 0.0001$. Statistical significance relative to the inactive L mutant sample was determined using ordinary one-way ANOVA with Dunnett's multiple comparison correction. **d** N- and C-terminal BioID2 tagged constructs were transfected into HEK293T cells, challenged with EBOV at an MOI of 1, and stained for BioID2 protein

(green) and EBOV VP30 as a marker of infection and inclusion bodies (red) at 20 hpi with cell nuclei stained with Hoechst 33342 (blue). Overlap is indicated by yellow. Individual channels for each image are shown in Supplementary Fig. 2a. One technical replicate of one representative biological replicate is shown. Scale bar = 25 μm. **e** BioID2-tagged VP40 incorporation into filamentous viral particles was evaluated by transfecting cells with each construct, which were then infected and stained as described above. Insets show magnification with BioID2 staining apparent at the plasma membrane, with budding filamentous and torus-shaped structures indicative of virus particles. Scale bar = 10 μm. One technical replicate of one representative biological replicate is shown. **f** Biotinylation of cellular proteins was measured in cell lysates after inducing expression of each indicated protein by tetracycline and adding biotin to cell medium. GFP-BioID2 and Flp-In TREx 293 cells not expressing any BioID2 construct were used as non-specific biotinylation controls (last two lanes). Blots were probed with fluorescently labeled streptavidin. One representative experimental replicate of each construct is shown. Biotinylation of all four replicates is shown in Supplementary Fig. 2b. Source data are provided as a Source Data File.

C-termini (Supplementary Data 1, Supplementary Fig. 2c). Fusions that had low expression levels and trVLP assay activity (VP30-NT and VP35-CT) or that disrupted localization (VP40-CT) yielded very few interactions. However, BioID2 fusions that had reduced activity in the trVLP assay still yielded known interactions, including well-characterized interactions between NP and PPP2R2A, and VP30 with SRPK2 and RBBP6. The interaction between VP30 and SRPK2 was only identified with the N-terminal fusion of BioID2 to VP30.

**Network analysis and viral protein mapping reveals potential interactions of virus proteins with host protein complexes.** We considered only those interactions identified by constructs that localized similarly to untagged proteins and/or showed activity in the minigenome assay for subsequent analyses, excluding interactions with L-NT, VP30-NT, VP35-CT, and VP40-CT BioID2 tagged constructs. As for other PPI screens, a hub and spoke network model was generated to show the pairwise relationships of identified virus-host protein interactions (Fig. 2a). However, many cellular proteins execute their functions as part of larger protein complexes, so we sought approaches to relate hits according to known host protein-protein interactions (PPI) that could enrich for the actions of protein complexes. Computationally connecting the BioID hits using known PPI interactions in the STRING database[31] yielded a highly interconnected mass of interactions involving 260 hits (Supplementary Fig. 3a). Alternatively using Metascape to map proteins, which uses the traditional edge-density MCODE algorithm[32] yielded subnetworks of protein complexes (Supplementary Fig. 3b). However, this edge-density filtering algorithm considers only the number of interactions in generating subnetworks, which led to a bias in highly connected proteins being sequestered in a large subnetwork of 42 proteins.

To better identify subnetworks of connected proteins and prioritize inclusion based on hit score and PPI interaction confidence, we implemented the Prize-Collecting Steiner Forest (PCSF) algorithm[33] which optimizes paths using cost-based evaluations of the connections. PCSF prioritizes paths with higher confidence (low cost) interactions from the underlying human PPI and negatively weights low-confidence (high cost) interactions (Fig. 2b). To apply PCSF to the BioID data set, we used the PCSF package in R and made this script available through Zenodo (see Methods). PCSF mapped protein hits by assigning prize values based on SAINT scores (Supplementary Data 5) onto an underlying interactome of human PPIs (Fig. 2c). We used the Human Integrated Protein-Protein Interaction rEference v2.3 (HIPPIE), an interactome assembled and scored from experimentally validated PPIs in which interactions are given a confidence metric (edge weight) based on the number of publications and types of experimental approaches supporting the interaction[34]. The optimal path connecting the highest prized hits in the dataset was then determined. In addition to considering the confidence of

interactions underlying different paths between the highest prized hits, PCSF also prioritized connecting as many hits as possible, using pathways that incorporated multiple lower prize hits, if necessary (Fig. 2b, upper route).

PCSF linked 335 hits with an average edge confidence of 0.80 (range 0.63 to 0.99), indicating a high confidence network (Supplementary Data 6). The virus proteins interacting with each host protein were then annotated onto the network (Fig. 2d, colored circles, Supplementary Data 7). While most interactions were between only one virus and host protein, 16% of hits were associated with multiple viral proteins (Fig. 2d, segmented circles). From this analysis, groups of host proteins related by a common virus protein(s) were apparent. To determine if these groups had functional relevance, the network was clustered based on network topology and gene list enrichment was performed on each cluster using EnrichR[35] (Fig. 2e). Interestingly, clusters of host proteins that had been biotinylated by the same virus protein showed strong enrichment (adjusted $p$-values 0.03 to $3.84 \times 10^{-18}$) for functional pathways and were annotated as an additional layer onto the network (Fig. 2e, Supplementary Data 8).

PCSF subnetwork clusters included known EBOV-host protein interactions but also revealed new associations with functionally related host proteins. For example, VP24 was associated with multiple karyopherin (KPNA) proteins[23], as well as complexes that regulate gene expression (nucleosome remodeling and deacetylase (NURD) and TAFIID complexes, Fig. 2e, clusters 2 and 5). Consistent with the known role of actin in EBOV transport and assembly[36,37] multiple regulators of actin dynamics were identified as partners of NP and VP24 (Fig. 2e, cluster 12). Furthermore, our analysis linked NFκB signaling to a cluster of NP- and VP35-associated host proteins, including MAP3K7 (TAK1), the MAP3K7-binding proteins TAB1, 2 and 3 proteins, and the MAP3K7 deubiquitinase, CYLD (Fig. 2e, cluster 15). Although the viral glycoprotein was previously found to induce NFκB signaling[38], these observations suggest additional viral factors modulate this pathway. Finally, NP labeled multiple components of the gamma-tubulin ring complex (g-TuRC), which functions as a template to nucleate microtubule formation (Fig. 2e, cluster 3)[39].

We focused on a cluster of five proteins, DCP1A, DCP1B, DCP2, EDC3, and EDC4, labeled by VP35 for more in depth analyses (Fig. 2d inset). This cluster was enriched for factors related to mRNA decay by 5' to 3' exoribonuclease and processing-body (P-body) formation (Fig. 2e, cluster 17). Consistent with the functional enrichment, all five proteins are components of the host mRNA decapping complex[40]. Since biotinylation can involve direct as well as indirect interactions within 10 nm of the BioID2 enzyme[18] we predicted this cluster of proteins represented VP35 interaction with an intact host protein complex.

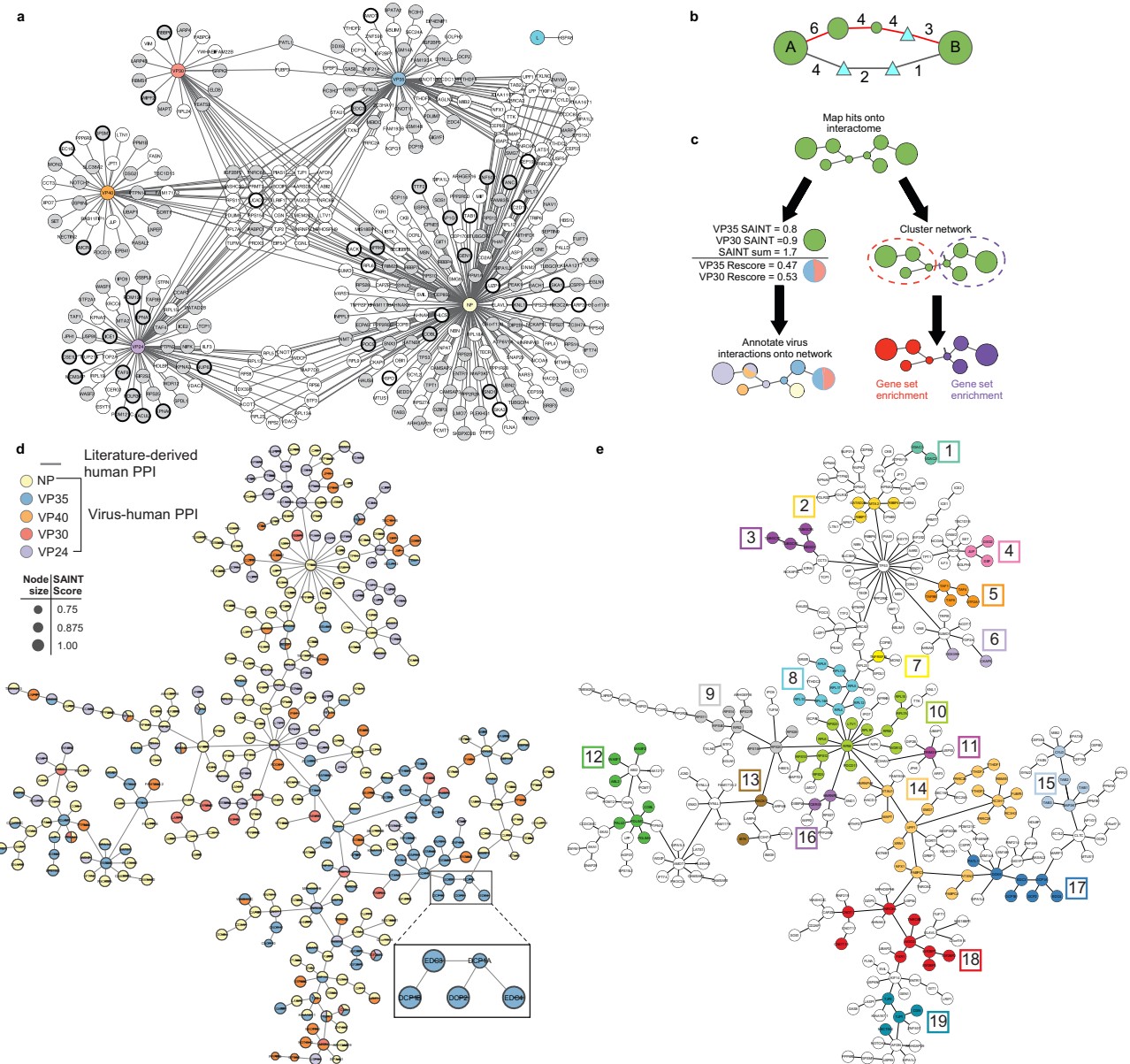

**Fig. 2 | Network analysis reveals virus interactions with putative host protein complexes. a** Hub-and-spoke network showing 441 high-confidence interactions from the EBOV BioID2 screen. Gray circles represent proteins identified as unique hits using both GFP and other viral proteins as controls. Circles with thicker borders are 64 proteins identified with both N- and C-terminal tagged constructs. **b** PCSF interrelates BioID hits with the HIPPIE human protein-protein interaction (PPI) network. The algorithm connects hits via shortest pathways defined by the PPI network, considering interaction confidence and the presence of other hits. Hits are shown as green circles, where node size reflects BioID score. Two paths are shown connecting scored hits A and B. The lower path is shorter but less supported by literature citations and requires adding two non-hit proteins (Steiner nodes, triangles). The upper path is longer, includes more screen hits, and has higher-confidence edges with more literature support. The algorithm preferentially includes the upper path in the final solution. **c** Left Rescoring of hits identified by more than one viral protein. Host proteins were partitioned according to each viral protein's contribution to the total SAINT score. Right Functional enrichment of subnetworks. Clusters identified by edge-betweenness (red and purple dashed circles), then subjected to gene set enrichment. **d** Full network generated by PCSF linking BioID2 hits through HIPPIE interactions. Colored circles indicate putative virus-host protein interactions. Lines connecting circles represent human PPIs. Five mRNA decapping complex proteins, each labeled by VP35, are highlighted in the inset. **e** Network annotated with pathways from Enrichr. Subnetworks are color-coded and numbered. Functional enrichments are in Supplementary Data 8. Source data are provided as a Source Data File.

## EBOV VP35 interacts with the host mRNA decapping complex scaffold protein EDC4

To identify which member of the mRNA decay complex was a direct binding partner for VP35, co-immunoprecipitations were performed from lysates of HEK293T cells transfected with plasmids encoding FLAG-tagged VP35 together with HA-tagged DCP1A, DCP1B, DCP2, EDC3 or EDC4. Under the conditions tested, VP35 only coprecipitated with EDC4 (Fig. 3a, last lane). To confirm the EDC4-VP35 interaction, pulldowns were performed with anti-FLAG antibody, with HA-tagged EDC4 found to coprecipitate with VP35 (Fig. 3b). These data were consistent with mass spectrometry data from the BioID screen, in which EDC4 was the most abundantly labeled decapping complex component in the VP35 BioID (Supplementary Fig. 4a). Overall, the findings indicate VP35 interacts with the decapping complex by binding EDC4.

To evaluate VP35-EDC4 interaction in EBOV infected cells, cells were stained with VP35 and EDC4 antibodies at 20 hours post infection

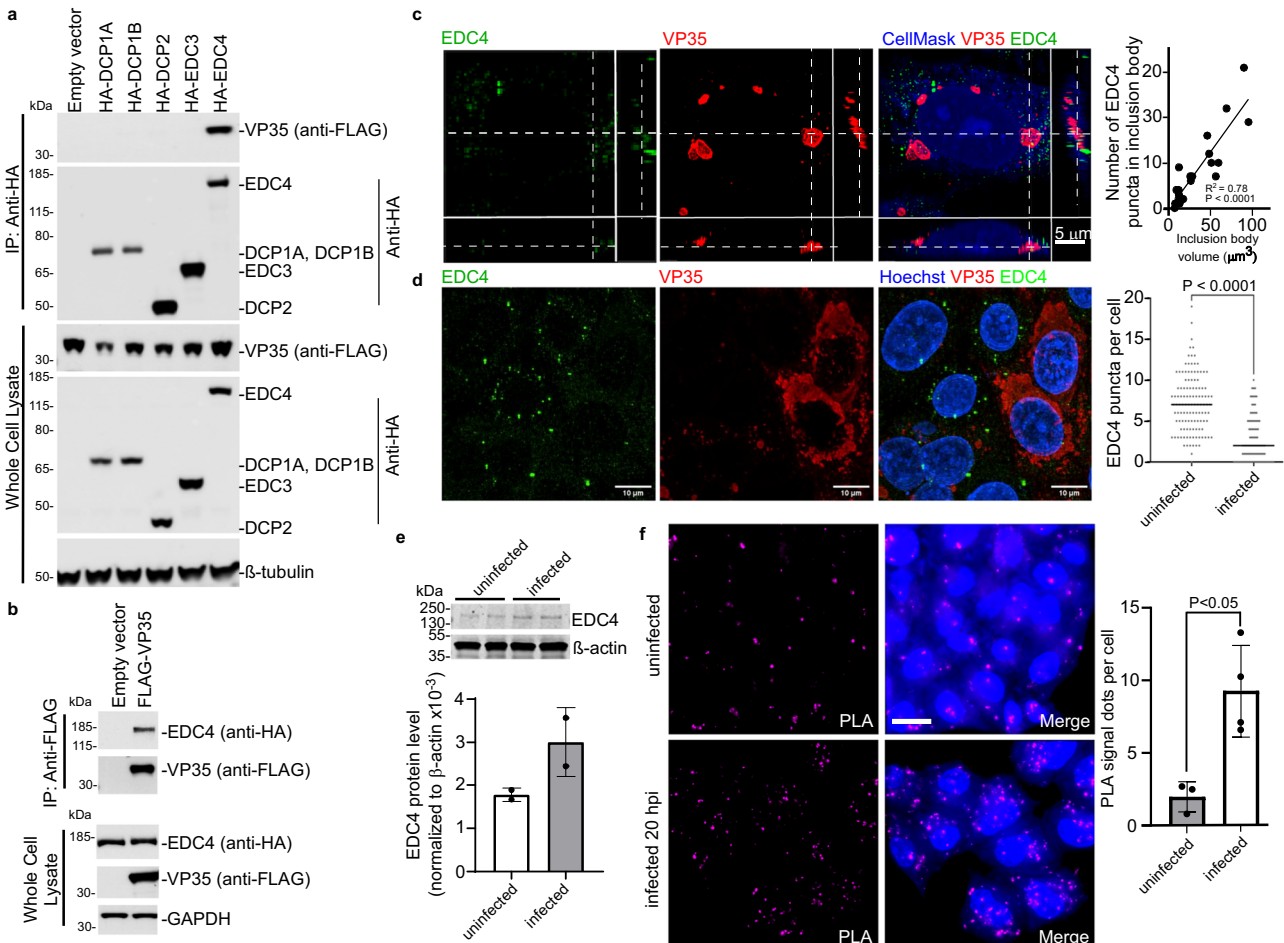

**Fig. 3 | VP35 interacts with the decapping complex through EDC4. a** Pulldowns of HA-tagged mRNA decapping proteins using anti-HA beads in the presence of FLAG-tagged VP35 show VP35 coprecipitates only with EDC4 (lane 6). One of two biological replicates is shown. **b** Confirmation of VP35-EDC4 interaction by pull-down of FLAG-tagged VP35 using anti-FLAG beads. One of two biological replicates is shown. **c** Cells challenged with EBOV at MOI 1 were fixed at 20 hpi, labeled with VP35 and EDC4 antibodies, and imaged by super-resolution confocal microscopy (48 nm resolution in x-y plane). Optical slicing of z-stacks shows x-y axis (top left), x-z and y-z axes (lower and right panels). EDC4 puncta (green) were present throughout the cell and within VP35-positive inclusion bodies (red), with colocalized puncta in yellow. Z-planes spanned 500 nm. Focus planes of x-z and y-z axes are shown with dashed lines. Merged image includes CellMask (blue), EDC4 (green), and VP35 (red). Scale bar = 5 μm. Right Quantification of EDC4 puncta within 21 inclusion bodies of varying volumes. Linear regression yielded $R^2 = 0.78$, $p = 9.51E$-08. One technical replicate of one representative biological replicate is shown.

**d** Cells were infected at MOI 0.5 to visualize both infected and uninfected cells. EDC4 puncta volumes were measured in VP35-positive infected ($n = 99$) and uninfected ($n = 119$) cells using Imaris software. Representative merged image and individual channels shown with quantification at right (unpaired t-test, $p < 0.0001$). **e** EDC4 levels in EBOV- or mock-infected cells were measured by immunoblot and quantitated relative to β-actin using image densitometry and area under the curve analysis. Values were scaled by $10^4$ to account for low signal intensities. One representative biological replicate containing two technical replicates is shown. **f** Proximity ligation assay (PLA) with antibodies to EDC4 and VP35. Cells were challenged with EBOV or mock-infected and fixed at 20 hpi. Left shows PLA signal (magenta); right shows PLA merged with CellMask (blue). Scale bar = 25 μm. Quantification shown at right (full images in Supplementary Fig. 4b). One representative biological replicate with four technical replicates is shown. Data are presented as mean ± SD. Statistical significance was determined using an unpaired t-test ($p = 0.0131$). Source data are provided as a Source Data File.

(hpi) and imaged using super-resolution confocal microscopy (Fig. 3c). While EDC4 was distributed throughout the cell cytoplasm as puncta, it was visible within VP35-containing inclusion bodies. Optical slicing through inclusions showed overlap of VP35 and EDC4 staining with the number of puncta being proportional to the size of the inclusions (Fig. 3c, right panel). Furthermore, we observed changes in the distribution and morphology of EDC4 staining patterns during infection. Uninfected cells had an average of 7 large EDC4 puncta (>0.1 um³) per cell, whereas EBOV infected cells had an average of 2 puncta (Fig. 3d). Since overall cellular EDC4 levels trended upwards but were not significantly different between uninfected and infected cells (Fig. 3e), the EDC4 in cells is therefore redistributed during infection. We further confirmed the association of each protein using proximity-ligation assays (PLA), which produce signal when proteins are within 40 nm of

each other. EBOV-infected cells displayed an average of 6 times more PLA puncta per cell compared to uninfected cells ($p < 0.05$, Fig. 3f, Supplementary Fig. 4b), indicating that VP35 and EDC4 are in close proximity. Taken together, these findings indicate that EDC4 binds to VP35, changes localization during EBOV infection and can enter inclusion bodies.

## EDC4 regulates EBOV replication at an early step in viral infection

Since EDC4 regulates mRNA stability through the decapping complex, we hypothesized that EDC4 depletion would affect virus RNA replication. The importance of the VP35-EDC4 interaction to the viral infection cycle was evaluated by depletion of EDC4 protein with two distinct siRNAs and then challenge with EBOV. Both siRNAs reduced EDC4

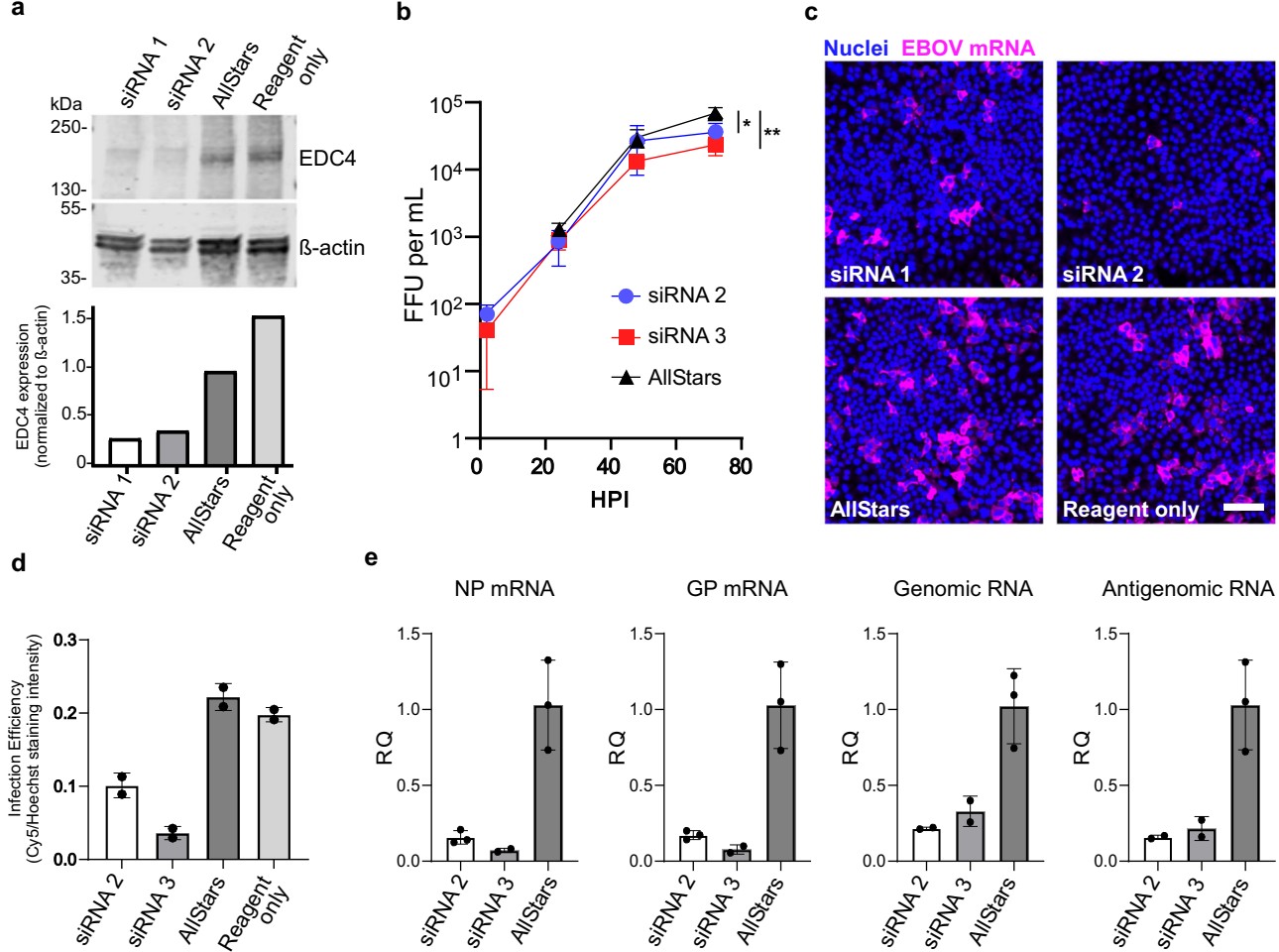

**Fig. 4 | EDC4 levels impact viral RNA production and EBOV growth kinetics.**
**a** EDC4 protein levels after siRNA treatment was measured by immunoblotting with EDC4-specific antibodies. Lower panel shows quantitation of a representative replicate of three independent experiments measured by area under the curve and normalized to β-actin expression levels. **b** EDC4 was depleted using the indicated siRNA and samples were harvested at 2, 6, 18, 24, 48, and 72 hpi. EBOV was titered by FFU assays performed on VeroE6 cells. Three technical replicates of one biological replicate are shown. Statistical significance was calculated by one-way ANOVA. Data are presented as mean ± SD. *P*-values = * 0.0271; ** 0.0051. **c** Viral RNA synthesis was measured by RNAFISH staining using multiple oligonucleotides complementary to EBOV NP and VP35 mRNA. Scale bar = 100 μm. **d** Quantification of RNAFISH signal from 10 images for which panel c is representative. Two technical replicates of one representative biological replicate are shown. **e** Measurement of viral RNA species by qPCR. NP and GP mRNA levels were measured after cDNA synthesis using poly-dT primers. Genomic (negative sense) and anti-genomic (positive sense) RNA were measured using a two-step reaction with specific primers (see methods). One representative biological replicate is shown per qPCR, each containing two or three technical replicates as indicated. Source data are provided as a Source Data File.

levels >80% (Fig. 4a). The impact of EDC4 depletion on viral replication kinetics was measured by the titer of EBOV produced over 72 hours from cells depleted of EDC4 using a focus forming assay. Even though 10-20% of EDC4 remained in cells, we observed a significant 2-3-fold decrease in virus production in cells treated with each siRNA compared to controls (Fig. 4b), suggesting that EDC4 promotes viral production. Since EDC4 is involved in RNA stability, the amount and subcellular localization of viral RNA were analyzed using RNA FISH. Depleting EDC4 via siRNA resulted in a 56% and 84% reduction in viral RNA signal (*p* < 0.01) compared to non-targeting (AllStars) siRNA controls, confirming a significant role for EDC4 in viral RNA replication (Fig. 4c, d). To determine which steps in viral replication were affected, we measured genomic and antigenomic RNA species as previously described[41] and viral mRNA encoding either NP or GP in cells treated with siRNA. In cells depleted of EDC4, NP transcript levels dropped by 7- and 14-fold for each respective siRNA (Fig. 4e, first panel) while GP transcript levels were reduced by 3- and 6-fold (Fig. 4e, second panel). In comparison, genomic RNA was decreased by 3- to 5-fold (Fig. 4e, third panel) and by 4-6 fold for antigenomic RNA (Fig. 4e, last panel).

These findings indicate that EDC4 is needed for synthesis of virus genome and messenger RNA. However, since replication of antigenomic and genomic RNA cycles between each form, it was not possible to determine if one was preferentially affected by loss of EDC4.

## The EDC4 C-terminal domain recruits VP35 and dominantly interferes with viral replication

EDC4 forms a scaffold on which DCP1A and DCP2, the minimal proteins required for mRNA decapping[42], assemble. EDC4 consists of an N-terminal WD40 domain, a flexible linker and a C-terminal alpha-helical region containing proximal and distal domains (Fig. 5a, top). DCP1A specifically binds EDC4 at the N-terminal WD40 domain, while DCP2 binds the proximal domain at the C-terminal end[42]. The helical distal domain, which is highly conserved, does not bind members of the decapping complex but adopts a compact seven helix-turn-helix bundle structure and plays a role in EDC4 localization to cytoplasmic puncta associated with P-bodies[43]. To identify what part of EDC4 interacts with VP35, HA-tagged EDC4 constructs encompassing the

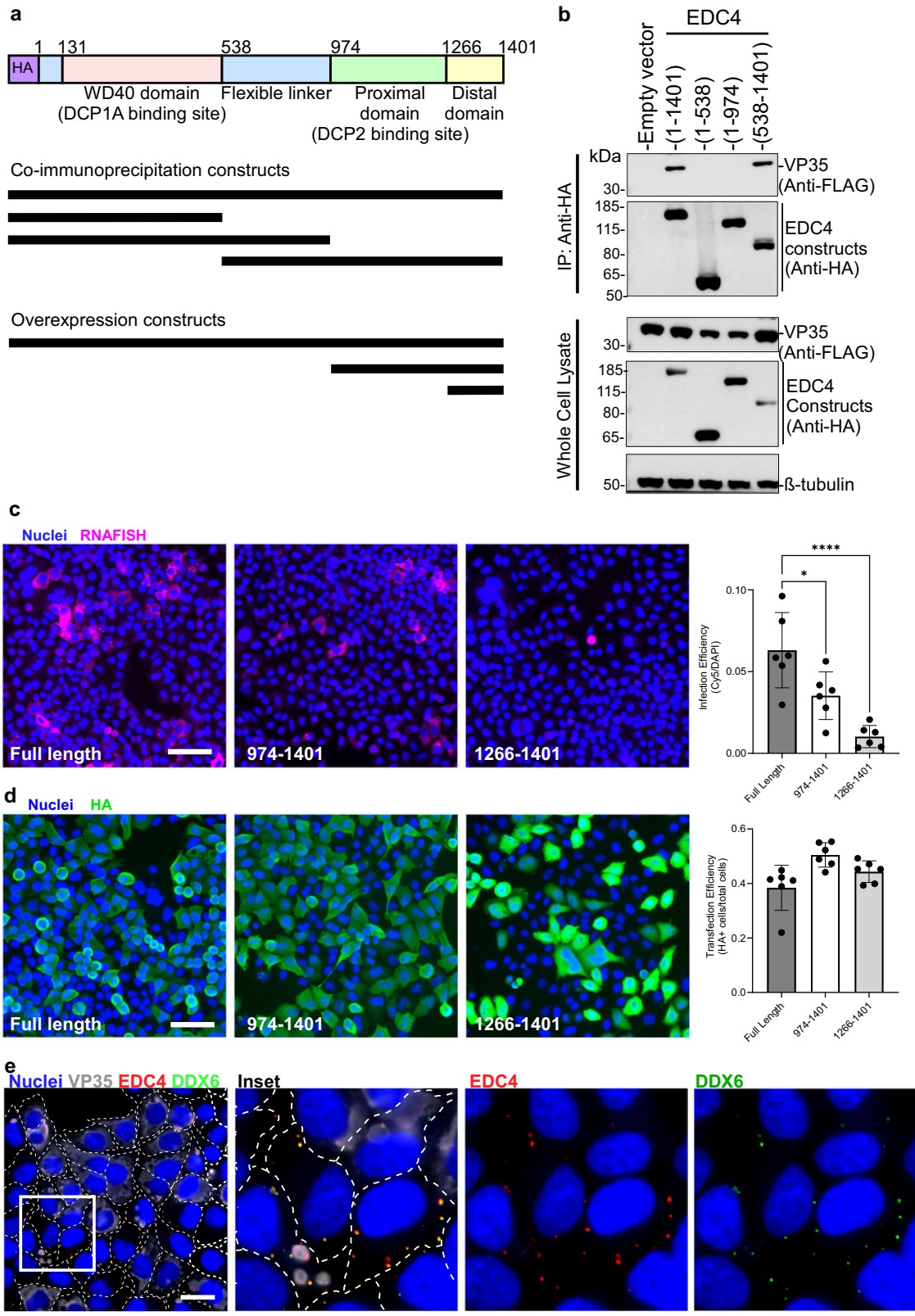

WD40 domain plus flexible linker or flexible linker domain plus C-terminal proximal-distal region of EDC4 were transfected into HEK293T cells with FLAG-tagged VP35 and co-immunoprecipitations performed (Fig. 5a, lower panel). VP35 strongly coimmunoprecipitated with full-length EDC4 (residues 1-1401) and a construct containing the flexible linker and the C-terminal alpha-helical domain (residues 538-1401) but was only weakly detected with the N-terminal WD40-containing domain and the flexible linker (residues 1-974), suggesting the binding site is within the C-terminal domain (Fig. 5b). We further tested whether overexpression of C-terminal subdomains affected viral replication in a dominant negative-like manner. HA-tagged full-length EDC4 and truncations containing the C-terminal proximal and distal domains or the distal domain alone (Fig. 5a, lower panel) were transfected into HeLa cells, which were then challenged with EBOV.

**Fig. 5 | EBOV VP35 interacts with the C-terminal domain of EDC4. a** Schematic of EDC4 showing binding sites for DPC1A and DCP2. In the lower panel, lines indicate constructs used in co-immunoprecipitation and overexpression experiments. **b** Identification of EDC4 subdomain responsible for VP35 interaction. The indicated HA-tagged EDC4 constructs were co-expressed with FLAG-tagged VP35, and immunoprecipitation was performed with anti-HA antibody. One representative replicate of two biological replicates is shown. **c** EBOV RNA levels were measured by RNAFISH (magenta) after overexpression of the indicated EDC4 domains. Cell nuclei were stained with Hoechst 33342 (blue). Statistical significance was calculated by one-way ANOVA with multiple comparisons. *P*-value = * 0.0178;

****$P$-value < 0.0001. Six technical replicates of one biological replicate are shown. Data are presented as mean $\pm$ SD. **d** Transfection efficiency of EDC4 constructs as determined by HA staining (green). Scale bars = 100 μm. The right panel shows the proportion of cells stained with HA antibody. Six replicates of one biological replicate are shown. Data are presented as mean $\pm$ SD. **e** Cells were challenged with EBOV and fixed 48 hpi, then stained with antibodies against VP35 (grey) to label infected cells, EDC4 (red), and DDX6 (green). Both EDC4 and DDX6 are markers of P-bodies. Cell boundaries are indicated by dashed lines. Scale bar = 25 μm. One technical replicate of a representative biological replicate is shown. Source data are provided as a Source Data File.

Cells were fixed at 20 hpi and viral RNA replication quantified by RNAFISH (Fig. 5c). Although expression of the proximal and distal domains moderately reduced infection, expression of the distal domain alone reduced infection 6-fold compared to full-length EDC4 (Fig. 5c). For each construct the frequency of transfected cells was similar though the distal domain (residues 1266-1401) was more highly expressed than full length or the proximal-distal (residues 974-1401) construct by immunofluorescence microscopy (Fig. 5d). Given that the distal domain of EDC4 plays an important role for recruitment to P-bodies, we then examined how infection affected the distribution of EDC4 and DDX6, which are commonly present in P-bodies[43]. Cells were challenged with EBOV and fixed 48 hpi to allow for multiple rounds of infection. In cells with distinct VP35-stained inclusion bodies, which represent earlier stages of infection, DDX6 and EDC4 puncta were visible in the cell cytoplasm and next to inclusion bodies, as seen in Fig. 3c. However, in cells with extensive VP35 staining, which indicates late-stage infection[19], the EDC4-DDX6 puncta were no longer detected (Fig. 5e). Overall, these observations show that VP35 interacts with the C-terminal half of EDC4, and overexpression of the distal domain disrupts infection likely through interaction with VP35. Higher level expression of VP35 seen in advanced stages of infection also caused loss of EDC4 puncta associated with DDX6, a marker of P-bodies, suggesting disruption of normal P-body organization.

To evaluate whether EBOV replication depends on decapping components other than EDC4, we tested the impact of knockdowns of DCP1A, B and DCP2 and accessory decapping protein EDC3 on infection efficiency. Unfortunately, we were unable to suppress DCP1A or B expression. However, cells treated with DCP2 and EDC3 targeting siRNAs showed strong reductions in protein levels (Fig. 6a). Depletion of DCP2, resulted in a 14-fold reduction in EBOV RNAFISH signal while depletion of EDC3 resulted in a 2-fold decrease (Fig. 6b, c). Like EDC4, DCP1A and EDC3 formed puncta in the cell cytoplasm with multiple puncta also present within VP35-stained inclusions (Fig. 6d, e). Larger inclusions harbored more puncta, particularly for DCP1A (Supplementary Fig. 5). Since DCP1A and DCP2 do not appear to directly bind VP35 (Fig. 3a), this suggests they either accumulate randomly or are recruited through EDC4 to inclusions. Overall, this work indicates that EBOV replication depends on multiple components of the decapping complex, which are likely recruited indirectly through EDC4.

## Discussion

We present a comprehensive protein proximity screen using all EBOV proteins except GP. Our screen identified many of the well-characterized EBOV-host cell protein interactions reported in the literature, provided additional support to less characterized interactions identified in other proteomic studies, and uncovered previously unidentified EBOV-host protein-protein interactions (Fig. 7). To facilitate analysis of this large dataset, we developed an approach to interrelate the virus-host interactome in the context of known human protein interactions. Annotating virus protein interactors onto this simplified network of host cell proteins revealed members of potential protein complexes and proteins associated with specific cellular functions interacting with the same viral proteins.

From this analysis, we identified proteins that comprise the mRNA decapping complex as important factors in EBOV replication and demonstrated that VP35 expression alters the expression pattern of this complex by binding to one of its members, the scaffold protein EDC4. We demonstrated that VP35 interacts with EDC4 in the same region as the enzyme DCP2, and showed that depletion of EDC4, DCP2, or EDC3 inhibited viral replication in a similar manner. Notably, our results indicate that VP35 interaction with the decapping complex is functionally significant, as loss of EDC4 led to a coordinated reduction in all viral RNA species, including genomic, antigenomic, and mRNA (Fig. 4e). Furthermore, expression of the EDC4 distal domain acted in a dominant-negative manner, impairing infection (Fig. 5c). These findings suggest a necessary role for EDC4 in viral RNA stability or processing, though further studies will be needed to dissect whether this occurs at the level of viral transcription, replication, or degradation.

While individual members of the decapping complex have previously been reported as interactors of EBOV proteins, our study provides evidence that the entire complex, rather than just EDC4, is involved in viral replication. Pichlmair et al. and Batra et al. identified EDC4 as an interactor of VP35 through affinity purification-mass spectrometry (AP-MS), but did not detect additional members of the complex, likely because conditions favored direct interactions over indirect associations[10,38]. Similarly, a split-TurboID screen detected EDC3 but failed to identify other complex members at significant abundance[16]. By leveraging proximity-dependent labeling coupled with network-guided analysis, we achieved a higher depth of coverage of host protein complexes, allowing us to identify functionally related interactors rather than isolated hits. This approach provides an advantage over conventional affinity purification studies, which often fail to capture transient or labile interactions.

The role of the mRNA decapping complex in viral replication remains an emerging area of study. In other viruses, components of the mRNA decapping complex have been implicated in both proviral and antiviral functions[27,44]. For example, Rift Valley Fever Virus, as part of cap snatching, competes with the decapping complex enzymatic subunit, DCP2, for cellular 5' methylated caps[45] effectively disrupting DCP2 acting as a virus restriction factor. Similarly, XRN1, which binds EDC4 within the same region as DCP2 and degrades decapped mRNAs[36], has been reported to have both a proviral and antiviral roles depending on the virus[46,47]. Although XRN1 was labeled by VP35 in our screen, it was mapped to a separate cluster of proteins associated with nonsense-mediated decay factors and so was not evaluated. Another protein that regulates mRNA stability, MARF1, was also identified as an interactor of both NP and VP35 but was mapped to an adjacent complex of trafficking proteins to the decapping complex. MARF1 is negatively regulated by EDC4[48] to stabilize cellular mRNAs. Given these diverse roles, the precise mechanism by which the decapping complex promotes EBOV replication remains unclear. It is possible that VP35 sequesters EDC4 and its associated factors to regulate viral RNA stability or facilitate viral transcription, though additional studies are needed to confirm these hypotheses.

The observed redistribution of EDC4 puncta in infected cells (Fig. 5e) raises additional questions about the interplay between viral

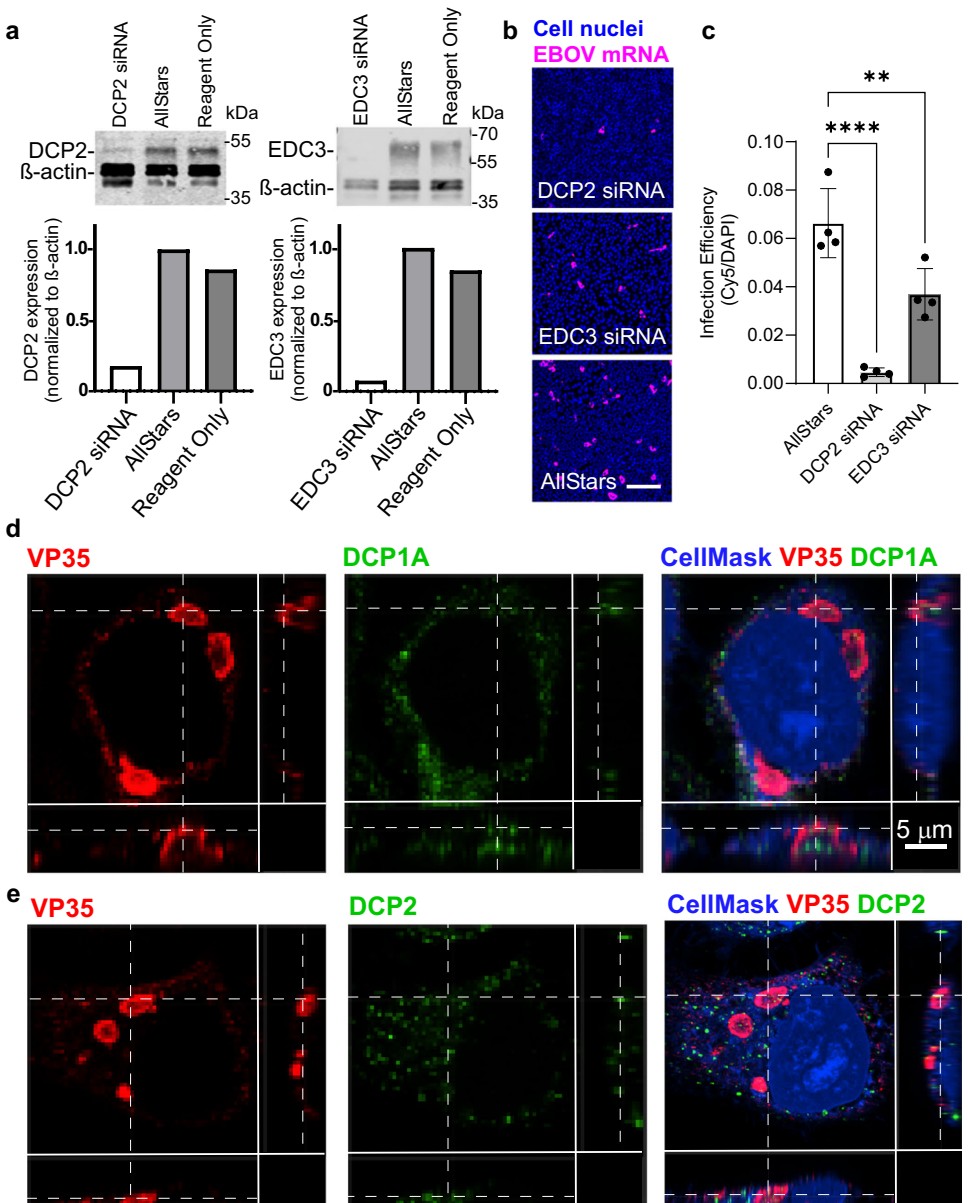

**Fig. 6 | EBOV replication depends on multiple components of the decapping complex. a** Depletion of DCP2 and EDC3 protein with siRNA. Corresponding bands on each blot (upper panels) were quantitated (lower panels). One of three biological replicates is shown. **b** Impact of depletion of DCP2 or EDC3 on EBOV infection measured through RNAFISH of viral mRNA. Samples were infected at an MOI of 0.8. Scale bar = 250 μm. **c** Quantification of b. Statistical analysis was performed using one-way ANOVA with multiple comparisons correction. Four technical replicates of one biological replicate are shown. Data are presented as mean ± SD. *P-value =

0.0111; **P-value = 0.0074; ****P-value < 0.0001. **d, e** DCP1A and DCP2 colocalize with VP35 during infection. Cells were challenged with EBOV at an MOI of 1, fixed at 20 hpi, stained with the indicated antibodies and imaged. Localization of **d** DCP1A and **e** DCP2 puncta in VP35 inclusion bodies was observed. Scale bars = 10 μm. Optical slicing of z-stacks were performed as described in Fig. 3 with the x-y axis at top left and x-z and y-z axes at lower and right panels and the focus of the x-z and y-z planes are indicated by dashed lines. One technical replicate of a representative biological replicate is shown. Source data are provided as a Source Data File.

replication and cellular RNA in cells. Members of the decapping complex have been associated with processing-bodies (P-bodies). These dynamic cytoplasmic structures are composed of protein and RNA and may be involved in RNA processing and/or mRNA sequestration. P-bodies fluctuate during replication of other viruses, such as SARS-CoV-2[49]. In our work, markers of P-bodies such as DDX6 and EDC4 formed several large puncta in the cytoplasm of uninfected cells that were replaced with smaller, more numerous puncta later in infection. Although the role of P-bodies in mRNA decapping is debated[50], the C-terminal region of EDC4 that interacts with VP35 is also responsible for targeting EDC4 to P-bodies and P-body assembly[37]. Several other major components of P-bodies associated with

decapping complex members were identified in our VP35 BioID screen in the network model, including DDX6, LSM14A, and LSM14B[50]. EDC3 interacts with the decapping complex by binding DCP1A and DCP1B in P-bodies[34,50]. While the role of EDC3 as an enhancer of mRNA decapping in yeast has been well characterized[50], its role in human mRNA decapping is understudied. Whether labeling of EDC3 in our screen is due to complex interactions between DCP1A and EDC4 or is an independent interaction of VP35 requires further study.

Our approach of combining proximity-dependent tagging with PPI-driven network modeling of virus-host interactions is broadly applicable to other viruses and cell types. Proximity tagging via biotinylation offers advantages over traditional pull-down mass

spectrometry approaches by capturing labile and transient complexes. However, once proteins are eluted for analysis, information on the specific protein partners forming these complexes is generally lost. Additionally, the expanded set of interactions requires further refinement before initiating labor-intensive mechanistic evaluations. To address this, we integrated known human protein-protein interactions into the virus-host interactome, aiming to recover biologically relevant groupings. Initial network visualization using STRING[25] resulted in an extremely dense and complex interaction map, providing limited guidance for follow-up studies (Supplementary Fig. 3). To better resolve functionally relevant clusters, we applied MCODE[51], a density-based clustering algorithm, via Metascape. This analysis successfully identified isolated protein complexes but components of well-characterized cellular pathways were largely disconnected from other hit proteins. For example, karyopherins, which function as nuclear import adapters, formed a distinct subnetwork with nodes related to actin cytoskeleton regulation (Supplementary Fig. 3)[52].

Given these limitations, we instead applied the Prize Collecting Steiner Forest (PCSF) algorithm, which optimizes connectivity between hit nodes while considering the confidence of known associations within the network. This approach generated an interconnected set of subnetworks without significantly reducing the number of mapped protein hits. In the case of karyopherins, this method revealed their connections to nucleoporin proteins NUP62 and NUP214, as well as POLR3C, an RNA polymerase III subunit that shuttles between the cytoplasm and nucleus, each linked via VP24 interactions. This refined network structure provided a more biologically meaningful framework for interpreting virus-host interactions and prioritizing mechanistic follow-up studies.

A key component of our analysis was the use of the Human Integrated Protein–Protein Interaction Reference (HIPPIE), a well curated PPI database based on experimental evidence present in the literature[28]. The combination of PCSF and HIPPIE yielded a greatly simplified, high stringency network that maintained cellular pathway flows. The extent of the produced network can be readily adjusted by tuning stringency parameters. A balance that yielded a network of 330 members was achieved by maintaining strict edge-confidence and degree parameters in conjunction with relaxed hit incorporation. The addition of an annotated information layer of virus protein interactions and gene-set enrichment for clusters provided further insight into impact on cell pathway function and directed us to evaluate the decapping complex and its interaction with VP35. In addition, other multi-protein complexes were seen, such as those involved in nuclear-cytoplasmic transport with VP24 or microtubule polymerization with NP, that will be studied in future work.

Despite the strengths of this study, several limitations are apparent. A common issue with BioID screening is that labelling of target proteins depends on the availability of lysine residues, so proteins without exposed lysines will be missed. Viral proteins expressed individually may miss interactions that occur with complexes of virus proteins. For this reason, using an intact virus is desirable but is difficult to perform due to constraints working safely with EBOV in a high containment laboratory. Second, our experiments were conducted in HEK293 cells, a model that supports EBOV replication but may not fully recapitulate the interactome of primary target cells. Additionally, tagging with BioID2 reduced activity of some proteins in the trVLP assay (VP30-NT and VP35-CT), which could impact their association with host factors. However, even when diminished in this assay, the tagged proteins still yielded validated hits, such as the interaction of VP30-NT with SRPK2, the kinase that phosphorylates VP30, and RBBP6, a well-characterized binding partner, supporting the robustness of the interaction dataset. Ultimately, all protein interaction assays have limitations, and multiple independent approaches are needed to characterize the entire EBOV interactome.

In conclusion, this study provides new insights into EBOV-host interactions, highlighting the decapping complex as a key regulator of viral replication. Our approach of combining proximity-dependent labeling with structured network analysis enabled the identification of entire functional complexes, rather than isolated interactors, and provided a mechanistic framework for understanding EBOV manipulation of host RNA metabolism. Future studies will focus on determining whether VP35 repurposes the decapping complex for viral RNA processing or interferes with its host functions. This work sets the stage for targeting RNA regulatory pathways as potential antiviral strategies against filoviruses.

## Methods

### Plasmids
Plasmids myc-BioID2-MCS and MCS-BioID2-HA were gifts from Kyle Roux (Addgene plasmid # 74223; http://n2t.net/addgene:74223; RRID:Addgene_74223; Addgene plasmid # 74224; http://n2t.net/addgene:74224; RRID:Addgene_74224). MYC-BioID2 and BioID2-HA were PCR amplified from mycBioid2 and MCS-BIOID2-HA and inserted into pCDNA5-FRT-TO (Invitrogen) to generate plasmids pCDNA5-FRT-TO-NT-BioID2 and pCDNA5-FRT-TO-CT-BioID2. EBOV genes and GFP were PCR amplified from existing plasmids and cloned into both NT- and CT-BioID2 plasmids. EDC4, EDC3, DCP1a, and DCP2 plasmids were obtained from GeneScript (Piscataway NJ) and cloned into a pCAGGS expression vector with the indicated epitope tag at the N-terminus. EDC4 and VP35 truncation mutants were made from the full length pCAGGS constructs. All plasmids were verified by sequencing.

### Cell lines and culture
Flp-In T-Rex 293 cells (Invitrogen, R78007) were maintained in DMEM supplemented with 10% FBS (Atlanta Biologicals Premium Select Fetal Bovine Serum, S11550) and 100 mg/mL zeocin (Gibco, R25001) at 37 °C. Stable Flp-In T-Rex 293 cells containing integrated BioID2 expression constructs were grown in DMEM supplemented with 10% FBS and 200 µg/ml hygromycin. All other cell lines were maintained in Dulbecco's Modified Eagle Medium (DMEM, Gibco™, Billings MT) supplemented with 10% fetal bovine serum (R&D Systems™, Minneapolis MN) at 37 °C with 5% $CO_2$. HeLa cells, VeroE6 cells and HEK293T cells were purchased from ATCC® (Manassas, VA).

### Development of BioID2 stable cell lines
Stable cell lines with the pCDNA5-BioID2 plasmids integrated were generated by co-transfecting Flp-In T-Rex 293 cells with a single BioID2 plasmid (100 ng) and either pOG44 Flp-Recombinase Expression Vector (Invitrogen, V600520, 1 µg) or pFLPo (1 µg, a gift from Philippe Soriano, Addgene plasmid # 13792; http://n2t.net/addgene:13792; RRID:Addgene_13792[45]) using Lipofectamine 2000 (Invitrogen, 11668019) according to the manufacturer's protocol. Beginning the day before transfection, cells were incubated in DMEM + 10% FBS without zeocin. On the day after transfection, the cells were transferred to a 10 cm tissue culture plate or a T75 flask. At 48 h post transfection, the medium was replaced with DMEM + 10% FBS + 200 µg/ml hygromycin. Cells were incubated until colonies appeared (-14 days), at which point the cells were passaged and transferred to T75 flasks, expanded, and frozen.

### Expression of the BioID2 fusion proteins
Flp-In T-Rex 293 cell lines containing BioID2 expression constructs were incubated in DMEM + 10% FBS + 1ug/mL tetracycline (no hygromycin) for 48 hours. Cells were dislodged from the plate by gentle pipetting with PBS, collected by centrifugation for 5 min at 500 x g, and lysed in 50 mM Tris pH 8, 5 mM EDTA, 150 mM NaCl, 1% Triton X-100.

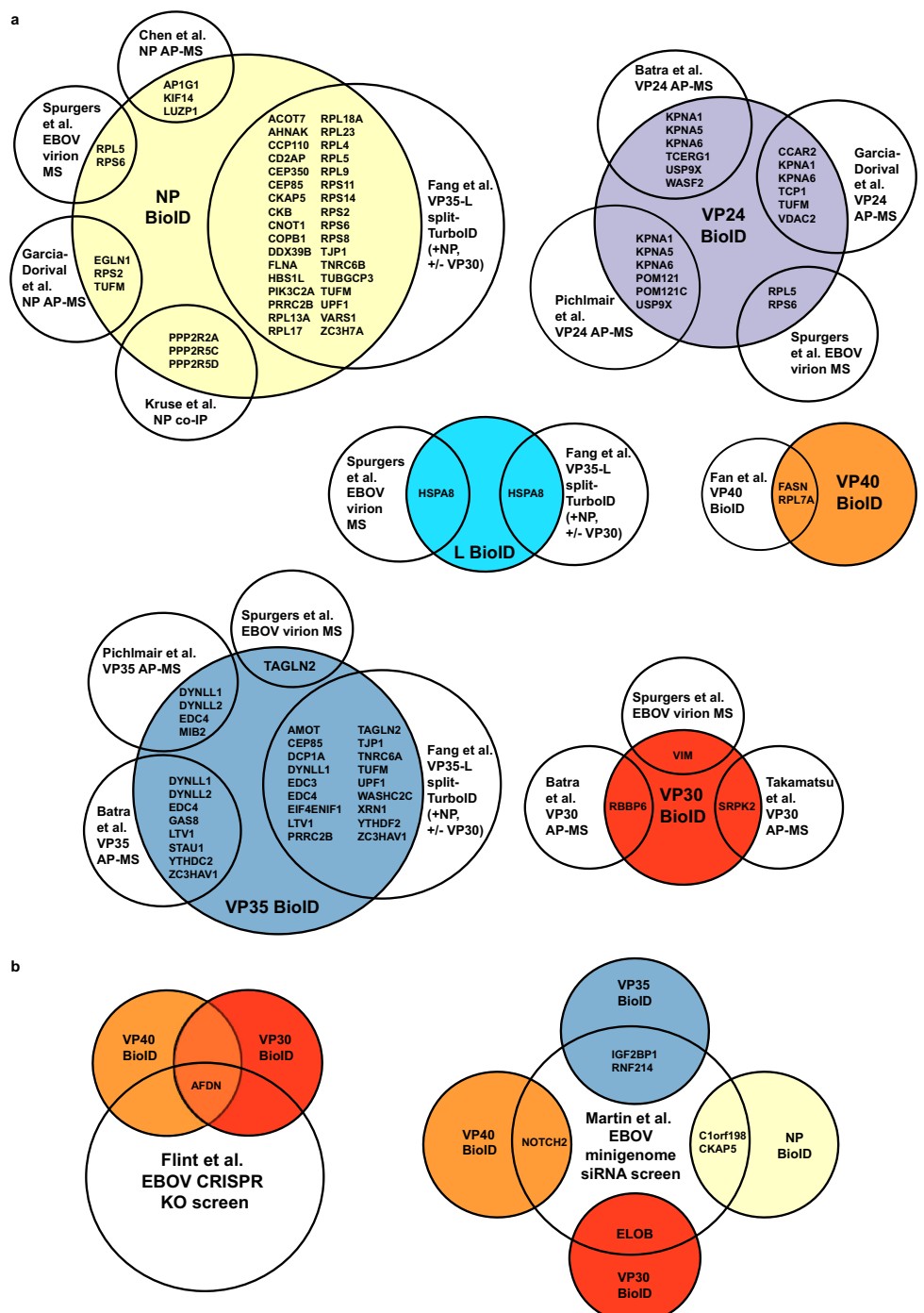

**Fig. 7 | Comparison of this BioID2 screen with other EBOV PPI and genetic screens. a** BioID hits for indicated virus proteins in this study (colored circles) were compared to other published EBOV PPI screens for those indicated proteins, including mass spectrometry of purified virus particles (Spurgers et al.[28]), affinity-pulldown mass spectrometry using NP, VP35, VP24, VP40 and VP30 as bait proteins to identify interactions (Garcia-Dorival et al.[13,14], Chen et al.[29], Pichlmair et al.[30], Batra et al.[10]), and other BioID screens performed using VP40 or the split TurboID L-

VP35 + /-VP30 complex to identify proximal interactors (Fan et al.[15], Fang et al.[16]). No overlap was identified with the NP affinity purification screen performed by Mor-witzer et al.[12]. **b** BioID hits were compared to genetic depletion screens performed for EBOV (Flint et al.[26], Martin et al.[27]). The EBOV entry screen performed by Carette et al.[44] was not included as it involved EBOV GP, which was not included in the present study.

Protein expression was quantitated through SDS-PAGE and western blotting.

**Biotinylation of cellular proteins**
Expression of BioID2 fusion proteins was induced by addition of tetracycline as described above. At 24 hours post addition, 50 mM

biotin (Sigma, B4639) was added to the culture medium, and the cells were incubated for another 24 hours, at which point cell pellets were collected. To confirm biotinylation, proteins were separated on SDS-PAGE gels and analyzed via western blotting. For mass spectrometry, cell pellets were flash frozen and stored at -80 °C.

## Purification and mass spectrometry analysis of biotinylated proteins

Biotinylated proteins were purified using streptavidin-sepharose beads[46,47]. Cell pellets from quadruplicate cultures of each BioID2 construct were thawed on ice and resuspended in 500 µl BioID lysis buffer (50 mM Tris (pH 7.5), 150 mM NaCl, 0.4% SDS, 1% IGEPAL, 1.5 mM MgCl2, 1 mM EGTA) supplemented with 250 U/ml Benzonase (MilliporeSigma, 712053) and protease inhibitors (Millipore Sigma, S8830). After one freeze-thaw cycle, samples were incubated for 30 minutes at 4 °C with rotation, subjected to centrifugation for 20 minutes at 21,130 x g at 4 °C. Supernatants were transferred to fresh tubes containing 35 µl washed streptavidin-sepharose beads (GE Healthcare) in BioID lysis buffer. Samples were incubated overnight at 4 °C with rotation. Beads were collected by centrifugation for 2 minutes at 500 x g and washed sequentially with BioID Lysis Buffer (1X), BioID Wash Buffer (50 mM Tris pH7.5, 2% SDS) (2X), BioID Lysis Buffer (2X), and 50 mM ammonium bicarbonate (pH 8.0) (3X). Beads were resuspended in 200 µl 50 mM ammonium bicarbonate (pH 8.0) and stored at -80 °C after removing 10 µl to confirm pull down of biotinylated proteins by western blotting. Samples were digested with trypsin and subjected to liquid chromatography-tandem mass spectrometry (LC-MS/MS) analysis using Dionex UltiMate 3000 RSLC nano System (Thermo Fisher Scientific, Waltham, MA, USA) coupled to the Q-Exactive High-Field (HF) Hybrid Quadrupole Orbitrap MS (Thermo Fisher Scientific, Waltham, MA, USA)[47]. The reverse phase peptide separation was achieved using a trap column (300 µm ID × 5 mm) packed with 5 µm 100 Å PepMap C18 medium, and then separated on a reverse phase column (50-cm long × 75 µm ID) packed with 2 µm 100 Å PepMap C18 silica (Thermo Fisher Scientific, Waltham, MA). The column temperature was maintained at 50 °C to maintain column pressure. The mobile phase solvent A was 0.1% Formic Acid (FA) in water and solvent B was 0.1% FA in 80% acetonitrile (ACN). The loading buffer was 98% water, 2% CAN and 0.1% FA. Peptides were separated by loading into the trap column in a loading buffer for 5-min at 5 µL/min flow rate and eluted from the analytical column at a flow rate of 300 nL/min using a 130-min LC gradient as follows: linear gradient of 5.1 to 27% of solvent B in 80 min, 27-45% in next 20 min, 45-100% of B in next 5 min at which point the gradient was held at 100% of B for 7 min before reverting back to 2% of B at 112 min, and hold at 2% of B for next 18 min for equilibration. The mass spectrometer was operated in positive ion and standard data-dependent acquisition mode with the Advanced Peak Detection function activated for the top 20n. The fragmentation of the precursor ion was accomplished by a collision energy setting of 27%. The resolution of Orbitrap mass analyzer was set to 120,000 and 15,000 for MS1 and MS2, respectively. The full scan MS1 spectra were collected in the mass range of 350-1,600 m/z, with an isolation window of 1.2 m/z and a fixed first mass of 100 m/z for MS2. The spray voltage was set at 2 and Automatic Gain Control (AGC) target of 4e5 for MS1 and 5e4 for MS2, respectively.

To identify biotinylated proteins, LC-MS/MS data were analyzed with Mascot Daemon with the following search parameters: Enzyme: Trypsin/P; Max missed cleavage:2; MS1 mass tolerance: 0.05 Da; MS/MS mass tolerance: 0.2 Da; Database: Human from Uniprot. Proteins identified with a false discovery rate (FDR) < 1% were submitted to analysis by SAINTexpress implemented in the Contaminant Repository for Affinity Purification (CRAPome) www.crapome.org website for analysis[21]. Proteins were considered high confidence hits if their Bayesian False Discovery Rate (BFDR) was <0.01 when compared to negative controls (GFP-BioID2 and parental Flp-In T-Rex 293 cells incubated in with tetracycline and biotin). To identify proteins that specifically interacted with a given viral protein, mass spectrometry data from a single BioID2-tagged EBOV protein (using both NT and CT samples) was analyzed in SAINTexpress[21] using BioID data from all other viral proteins plus GFP-BioID2 and parental Flp-In T-Rex 293 cells as negative controls.

For this more stringent analysis, we considered a protein to be a high-confidence hit if its BFDR was <0.05.

## Transcription- and replication-competent virus-like particle (trVLP) assays

TrVLP assays were performed in triplicate in 6-well format[48]. HEK 293 T/17 (ATCC CRL-11268) cells were seeded at 0.45 × 10^6 cells per well. On the following day when the cells reached ~ 50% to 60% confluency, cells were transfected with pCAGGS expression plasmids encoding EBOV NP (125 ng), VP35 (125 ng), VP30 (75 ng), and L (1000 ng), or the respective NT-BioID2 or CT-BioID2 fused constructs, plus expression plasmids for T7-polymerase (125 ng) and pT7.1-4cis-vRNA-EBOV-GP-nluc (250 ng) using 6 µL of TransIT-VirusGEN Transfection Reagent (Mirus Bio LLC, Madison, WI, USA, MIR 6703) following the manufacturer's instructions[49,50]. pCAGGS-FF-luc2 (1 ng; expresses firefly luciferase) was included as a transfection control. pCAGGS-L-AAA, in which the active site residues D742, N743 and Q745 were mutated to alanine, was used as a negative control[51].

At 48 h after the transfection, cells were trypsinized and resuspended in 1 ml of DMEM supplemented with 10% fetal calf serum (FCS). Luciferase assays were performed using the Nano-Glo Dual-Luciferase Reporter Assay System (Promega Corporation, N1610) in opaque 96-well plates with 20 µL of resuspended cells. Luminescence was measured using a Synergy Neo2 Reader (BioTek). NanoLuc luciferase activities were divided by firefly luciferase values to correct for transfection efficiency and normalized to the sample with untagged EBOV genes, which was set to 1.

## Comparison to other EBOV protein and genetic screens

Using a threshold of BFDR < 0.05, an extended hit list was identified from the BioID screening. This list was compared to hit lists reported by other groups, assembled from Supplementary Data according to their published criteria for hit selection.

## Network analysis and gene ontology enrichment analysis

Proteins that were identified more than once (by N- and C- terminal tagged viral proteins) were represented as one host-virus protein pairing to eliminate duplicate host-virus interactions. These proteins were then assembled into an edge list and mapped using Cytoscape 3.10.0[52].

STRING analysis was conducted using STRINGv12.0[31] with the physical subnetwork, with score requirements set to a confidence threshold of 0.4, FDR stringency set to <0.05 and all default associations. The resulting network was exported to Cytoscape and genes were assigned a uniform color scale. Metascape analysis was conducted using the online software Metascape (http://metascape.org) tool with default parameters[32], and networks were annotated in R as described below.

Host protein hits identified in BioID screening were mapped onto the Human Integrated Protein-Protein Interaction rEference v2.3 (HIPPIE) through the PCSF package[33] in R version 4.3.1. SAINT scores for hits identified in either N-terminal or C-terminal BioID screens were assigned as prize values. Where 2 SAINT scores were present, due to the protein being a hit in both GFP or unique for specific virus proteins threshold criteria, the highest score was used. Three genes, with Uniprot IDs A0A024QYV8, G9K388, Q96PV7 were not present in the HIPPIE database and so were omitted from the analysis. The optimal network solution was developed with ω, β, and µ parameters set to values of 10, 2, and 0.01, respectively. 92.62% of hits reported in screening were also present in the HIPPIE interactome. Though functional enrichment was an optional application as part of the PCSF package, this portion of the package is now deprecated. To recapitulate the functional enrichment application of the PCSF package, the network resulting from application of the PCSF algorithm was clustered according to network topology using the edge betweenness community analysis algorithm in iGraph[53].

Functional enrichment analysis was performed on clusters using EnrichR through the EnrichR API package in R[35]. Functional enrichment databases were set to Gene Ontology (GO) Molecular Function 2023, GO Cellular Component 2023, GO Biological Process 2023, KEGG 2021 Human, and Reactome 2022. All R scripts written for analysis are available on Zenodo (10.5281/zenodo.13145450), and all networks are available on NDExBio (https://www.ndexbio.org/#/networkset/68b8747e-4ee9-11ef-a7fd-005056ae23aa?accesskey=35c17994cd78000521aeaa3078a63d7441dc365265d1c9e1008b61e41577cea2).

### Viral protein network annotation

Host proteins identified in BioID screening were scored for interactions with each of the six EBOV proteins tested. If no interaction was identified between a given host protein and viral protein, a score of 0 was assigned to the host protein score. If an interaction was identified between a given host and viral protein, an initial interaction score was identified from the SAINT score reported from mass-spectrometry analysis. If a host protein was identified to interact with only 1 viral protein, the sole SAINT score was converted to a value of 1. For host proteins found to interact with multiple viral proteins, SAINT scores for all host-virus protein interactions were summed, and each individual viral protein interaction SAINT score was then rescored as a fraction of the combined saint score. Final viral protein attribute scores were assembled using the spread function in the tidyr package[54] in R and are reported in Supplementary Data 8. Scores were then assigned as network attributes using the iGraph package[53] in R, and the final attribute-mapped network was exported to Cytoscape[52] using the Rcy3 package[55] in R.

### Antibodies

Anti-DCP1A (clone D6VR1, catalog number 15365, lot #1), anti-EDC4 (catalog number 2548S, lot #2) and anti-HA (clone C29F4, catalog number 3724S, lot #11) antibodies were purchased from Cell Signaling Technology (Danvers MA) and were used at 1:200, 1:100, and 1:1000 dilutions for immunofluorescence, respectively. Anti-DCP2 antibody was developed by Prestige Antibodies (catalog number HPA057676, lot #R80862) and purchased from Sigma-Aldrich (St. Louis MO), and used at a 1:100 dilution for immunofluorescence studies and a 1:500 for immunoblotting. Anti-DDX6 antibody (catalog number PA5-18478, lot #WI3373004) was developed by Thermofisher (PA5-18478) and used at 1:100 dilution for immunofluorescence. Anti-BioID2 antibody (catalog number SS 3A5-E2, lot number 1014257-6) was purchased from Abcam (Cambridge, United Kingdom) and used at a 1:200 dilution for immunofluorescence. For immunoblotting, anti-EDC4 (catalog number 17737-1-AP) was purchased from Proteintech (Rosemont, IL) and used at a 1:1000 dilution. Anti-EDC3 antibody (catalog number A13763, lot #0065350201) was purchased from ABClonal (Woburn, MA) and used at a 1:500 dilution. Anti-β-actin antibodies were purchased from R&D Systems (catalog number MAB8929, clone 937215, lot #CJQV032104) and from Sigma Aldrich (Saint Louis, MO; catalog number A1978, clone AC-15, lot number 088M4804V) and were used at a 1:3000 dilution and 1:20,000 dilution, respectively, for immunoblotting. Anti HSP60 antibody (catalog number AF1800, lot #KRO0921121) was purchased from R&D systems and used at a 1:4000 dilution for immunoblotting. Anti-BioID2 antibody was purchased from BioFront Technologies (Tallahassee, FA) for immunoblotting (catalog number BioBID-CP-100, lot number 03290116). and used at a 1:40,000 dilution. Anti-EBOV VP35 antibody was developed and obtained from Dr. Daisy Leung (Washington University, St. Louis, MO) and used at a 1:1000 dilution for immunofluorescence and a 1:500 dilution for Western blotting. Alexa-Fluor 546 goat anti-mouse (catalog number A-11030, lot number 2026145) and Alexa-Fluor 488 goat anti-rabbit (catalog number A-11008, lot number 2500542) antibodies as well as Alexa-Fluor 488 chicken anti-goat (catalog number A-21467, lot

number 2328940), Alexa-Fluor 546 donkey anti-rabbit (catalog A10040, lot number 2411581) and Alexa-Fluor 647 donkey anti-mouse (catalog number A-31571, lot number2555690) antibodies were purchased from Invitrogen (Thermofisher Scientific, Waltham MA) and used at 1:1500 dilutions for immunofluorescence. Chicken anti-BioID2 primary antibody was purchased from BioFront Technologies (BioBID-CP-100). IRDye 800CW donkey anti-chicken secondary antibody (catalog number 925-32218, lot numbers C70201-05, C90213-15), IRDye® 800CW Goat anti-Rabbit IgG Secondary Antibody (catalog number 925-32211, lot number D01110-09) 1:10000), and IRDye® 680RD Goat anti-Mouse IgG Secondary Antibody (catalog number 926-68070, lot number D10217-01) were purchases from LI-COR Biotechnology (Lincoln, NE) and used at 1:10,000 as secondary antibodies for all relevant immunoblotting experiments.

### Virus propagation and infection

All EBOV experiments were performed under biosafety-level 4 (BSL4) conditions at the National Emerging Infectious Diseases Laboratories BSL4 suite in Boston, Massachusetts USA. Zaire EBOV was originally provided by Heinz Feldmann (Rocky Mountain Laboratories, Hamilton MT) and cultured on VeroE6 cells. Supernatants were collected 5 days post-infection after the development of cytopathic effect and clarified via centrifugation. Virus was then concentrated over a 20% sucrose cushion via ultracentrifugation. Titer was calculated by serially diluting the virus on HeLa cells and staining for virus via RNAFISH (see below) at 20 hpi. Unless otherwise indicated, cells were infected at an MOI of 2 for all experiments. Treatment with transfection reagent for siRNA and EDC4-overexpression experiments lowered the resulting cell susceptibility to EBOV.

For multi-step growth curve experiments, cells were infected at an MOI of 1. After one hour, input virus was removed and cells were washed with DMEM + 10% FBS. Supernatant was harvested at the indicated timepoints and frozen at -80 °C. Supernatants were then titered by serially diluting virus on VeroE6 cells and staining for EBOV GP. The number of GP foci were calculated for each sample.

### RNAFISH

Samples were fixed in 10% neutral buffered formalin for at least 6 hours to inactivate EBOV. EBOV mRNA staining was adapted from previously published methods[56]. Probes targeting NP and VP35 positive-sense mRNA and containing an additional 20 nucleotide flap sequence were designed (see Supplementary Data 9) and synthesized by Integrated DNA Technologies (IDT, Coralville IA). A secondary probe complimentary to the 20-nucleotide flap sequence (see Supplementary Data 9) was synthesized with Cy5 conjugates on both 5' and 3' ends by Genewiz Inc. (Azenta Life Sciences, Burlington MA).

### Immunofluorescence and Proximity Ligation Analysis

Samples were fixed in 4% paraformaldehyde or 10% formalin for at least 6 hours to inactivate EBOV. Samples were then washed in phosphate-buffered saline (PBS), permeabilized in 62.5 ng/mL digitonin, and blocked in 3.5% bovine serum albumin (BSA). Primary antibodies were diluted in 3.5% BSA and incubated at 4 °C overnight. Samples were washed in PBS and incubated with secondary antibodies diluted in 3.5% BSA for 1 hour at room temperature. Cell nuclei were stained with Hoechst 33342 (Thermofisher Scientific, Waltham MA) diluted in PBS. PLA was performed using the EDC4 and VP35 antibodies described above with the Duolink In-Situ Detection Red Kit (Millipore Sigma, Burlington MA) according to the manufacturer's protocol.

### Co-immunoprecipitations

HEK293T cells in 6 well plates were transfected with the indicated plasmids typically at 0.5 µg (HA-DCP1a, HA-DCP1b, HA-DCP2, HA-EDC3, HA-EDC4, HA-EDC4 1-974, HA-EDC4 538-1404) and Flag-VP35 at 1 µg by using Lipofectamine 2000 according to the manufacturer's

instructions (Invitrogen #11668500). After 48 h incubation, the cells were washed once with PBS, lysed in 240 µl NP40 lysis buffer (50 mM Tris HCl pH 8, 280 mM NaCl, 0.2 mM EDTA, 2 mM EGTA 10% glycerol, 0.5% NP40) and centrifuged at 16000 x g for 10 min at 4 °C. The whole cell extracts were incubated with 20 µl anti HA magnetic beads (Pierce™ Anti-HA Magnetic Beads #88836) or anti FLAG M2 beads (Sigma-Aldrich) for 45 minutes. A small portion of supernatant (40 µl) was saved as input. The beads were washed five times with NP40 lysis buffer. The IP complex was eluted in 50 µl NP40 buffer containing 1X SDS loading buffer (Thermo- scientific #J61337-AC) and heated to 95 °C for 5 minutes. Proteins were separated by SDS-PAGE gels (Invitrogen™ Bolt™ 4 to 12%, Bis-Tris #NW04120BOX) and subsequently transferred onto a PVDF membrane (Sigma-Aldrich #03010040001). The membranes were blocked with 5% blocking buffer (#1706404) for 1h at room temperature, followed by overnight incubation with anti-Flag (1:2500, Sigma-Aldrich #F7425) and anti-HA (1:2500, Invitrogen #71-5500) antibodies. The membranes were then incubated with secondary antibody (1:7500, Cell Signaling Technology #7074S), washed 3 times for 10 min each using 1X PBST and imaged with Western Chemiluminescent HRP substrate (PerkinElmer #NEL104001EA) in a ChemiDoc™ MP Imaging System (Bio-Rad).

## siRNA and plasmid transfections

Scramble control siRNA and siRNAs targeting EDC4 (catalog number 1027418, GeneGlobe IDs: SI04355365, SI05003345), EDC3 (catalog number 1027418, GeneGlobe ID: SI04359194), and DCP2 (catalog number 1027418, GeneGlobe ID: SI04211669) were purchased from Qiagen (Hilden, Germany). HeLa cells were seeded at 6000 cells per well in 96-well dishes. siRNAs were transfected at a concentration of 5 nM (EDC4 siRNA) or 10 nM (EDC3 and DCP2) siRNA per 96 well using 0.3 µL of Lipofectamine RNAiMax transfection reagent (Thermofisher Scientific, Waltham, MA). At 24 hours post transfection, cells were washed in growth media and infected with Zaire EBOV. For EDC4 expression experiments, full-length and truncated EDC4 plasmids were transfected into HeLa cells at 0.1 µg per 96 well using TransIT-LT1 (MirusBio, Madison, WI) according to the manufacturer's protocol. Twenty-four hours post transfection, cells were challenged with Zaire-EBOV, then inactivated 20 hpi with 10% neutral-buffered formalin. For localization experiments with BioID2-tagged viral protein constructs, plasmids encoding each construct were transfected into HeLa cells at 1 µg per chamber well of an 8-chamber well slides using TransIT-LT1 (MirusBio, Madison, WI) according to the manufacturer's protocol. Twenty-four hours post infection, cells were challenged with Zaire-EBOV.

## Microscopy and image quantification

For EDC4 knockdown experiments in which RNAFISH was used as a metric of viral replication efficiency as well as EDC4 truncation expression experiments, plates were imaged on a BioTek Cytation 1 (Agilent Technologies, Lexington MA) using a 10x objective and Cy5 and DAPI filters. Each well was imaged by collecting 16 non-overlapping tiled images. For DCP2 and EDC4 knockdown experiments, plates were imaged on a Nikon Ti2 fluorescent microscope (Nikon, Tokyo, Japan) using a 10x objective and 385 and 637 filters. For all decapping component knockdown experiments, infection efficiency was quantified by measuring the area occupied in µm² by 637 and 385 signals corresponding to RNAFISH and cell nuclei, respectively, in ImageJ or CellProfiler[57]. RNAFISH area occupied, and nuclei area occupied measurements were summed for each well, and total RNAFISH values were normalized to total nuclei area occupied measurements by division to generate infection efficiency. For infection experiments examining EDC4 and DDX6 distribution, images were taken on a Nikon Ti2 fluorescent microscope (Nikon, Tokyo, Japan) using a 20x objective and 385, 490, 550, and 637 filters. For PLA experiments, single plane images were taken using a 20x objective, while Z-stack images were taken using a 100x oil-immersion objective

with 2 µm separating each z-slice and 385 and 637 filters on a Nikon Ti2 fluorescent microscope (Nikon, Tokyo Japan). Images were deconvolved using Microvolution (Microvolution LLC, Cupertino, CA) in ImageJ and processed to give maximum intensity projections. For colocalization experiments between VP35 and decapping components, Z-stack images were taken using a 100x oil-immersion objective with 2 µm separating each z-slice and 385, 488, and 546 filters on a Nikon Ti2 fluorescent microscope (Nikon, Tokyo, Japan). Images were deconvolved using Microvolution (Microvolution LLC, Cupertino, CA). For super resolution microscopy experiments examining colocalization between VP35 and decapping components, z-stack images were taken on Nikon Spatial Array Confocal (NSPARC) Ti2 microscope using a 60x oil immersion objective and 561, 488, and 405 lasers, with 0.5 µm separating each z-slice, and deconvolved using NIS Elements Deconvolution software (Nikon, Tokyo, Japan). Images were then modeled using Imaris (Oxford Instruments, Concord, MA). VP35 and decapping component signals were thresholded, with VP35 modeled as a surface and decapping component puncta modeled as spots. To calculate puncta per inclusion body, inclusion bodies were filtered through volumetric filtering in Imaris, and decapping puncta within 0 µm of the filtered VP35 surfaces were indicated and counted.

## SDS-PAGE and Western Blotting

To analyze BioID2 fusion protein expression, samples were lysed as described above, combined with 2X SDS sample buffer, boiled, and separated on an SDS-PAGE gel. Samples were transferred to nitrocellulose and sequentially probed with primary (BioFront Technologies, BID2-CP-100, 1:20,000 dilution) and IRDye 800CW Donkey anti-Chicken Secondary Antibody (LI-COR Biotechnology, 925-32218). To confirm biotinylation in BioID screens, samples were lysed as described above and transferred to nitrocellulose and probed with IRDye 800CW Streptavidin (LI-COR Biotechnology, 926-32230, 1:3000 dilution). For mRNA decapping component KD and viral protein quantitation, Samples were lysed with RIPA buffer supplemented with Pierce Protease and Phosphatase Inhibitor (Thermofisher Scientific, Waltham, MA) and boiled in 1% SDS at 100 °C for 10 minutes to denature samples and inactivate EBOV. Samples were clarified via centrifugation, combined with Laemmli buffer, and loaded onto 4-20% polyacrylamide Mini-PROTEAN® TGX gels (BioRad, Hercules, CA). Following electrophoresis, samples were rapidly transferred to nitrocellulose membranes using the Trans-Blot Turbo transfer system (BioRad, Hercules, CA) and blocked in Intercept (PBS) blocking buffer (LI-COR Biosciences, Lincoln, NE) at room temperature for 1hour with rocking. Antibodies were diluted in Tris-buffered saline-tween (TBST) and membranes were incubated overnight at 4 °C with rocking. Membranes were washed twice in TBST, then incubated with fluorescent secondary antibodies diluted in blocking buffer for 1 hour. Membranes were then washed three times in TBST. Target protein quantities were normalized to housekeeping protein quantities. All blots were imaged on LI-COR Odyssey CLx or LI-COR Odyssey Imaging Systems and quantified in ImageStudio Lite (LI-COR Biosciences, Lincoln NE).

## PCR, Reverse-transcription PCR (RT-PCR), and Quantitative PCR (qPCR)

All decapping component KD samples infected with EBOV were lysed in TRIzol Reagent (Thermofisher Scientific, Waltham MA), and RNA extractions were performed via chloroform precipitation. RT-PCR was performed using the PrimeScript RT Reagent Kit (Takara Bio Inc., Kusatsu, Shiga, Japan), using oligo-dT primers to amplify mRNA, random hexamers to amplify non-capped and non-poly-adenylated RNAs, or EBOV-genomic specific primers. Quantitative PCR detecting viral mRNA was performed using the Luna Universal Probe qPCR

Master Mix (New England Biolabs Inc, Ipswich MA). Quantitative PCR detecting antigenomic and genomic EBOV strands was performed using the iTaq Universal SYBR Green Supermix (BioRad, Hercules CA). Primers and probes used to detect viral mRNA were designed using the IDT PrimerQuest design tool for PrimeTime qPCR assays. NP primer and probe sequences are as follows: 5′-ACTCCAT-CACGCTTCTTGAC-3′ (forward primer), 5′-CATGCGTACCAGGGA-GATTAC-3′ (reverse primer) and 5′-/56-FAM/TCAAGTATT/ZEN/TGGAAGGGCACGGGT/IABkFQ/-3′ (probe). GP primer and probe sequences are as follows: 5′-CAGTCCGGTCCCAGAATGTG-3′ (forward primer), 5′-TTTTCAATCCTCAACCGTAAGGC-3′ (reverse primer), and 5′-/56-FAM/-CATGTGCCG/ZEN/CCCCATCGCTGC/IABkFQ/-3′ (probe). 7SK snRNA was used as a housekeeping gene, and was quantitated using assay Hs.PT.58.38798776.g from IDT. Primers to detect EBOV genomic and antigenomic strands have been previously described[41].

### Statistical analysis and reproducibility
GraphPad Prism 10 for Windows (GraphPad Software, San Diego, CA) was used to perform one-way ANOVAs, unpaired t-tests, and simple linear regression. Significance was determined as any p-value < 0.05 with multiple comparisons correction. All experiments were performed with multiple replicates.

### Reporting summary
Further information on research design is available in the Nature Portfolio Reporting Summary linked to this article.

## Data availability
The raw mass spectrometry proteomics data generated in this study have been deposited in the MassIVE database under accession code MSV000093669. The R scripts used for network analysis and Cell-Profiler pipelines are available on Zenodo under the https://doi.org/10.5281/zenodo.13145450 and https://doi.org/10.5281/zenodo.16618585. The processed protein-protein interaction networks generated in this study are available on NDEx at https://www.ndexbio.org/#/networkset/68b8747e-4ee9-11ef-a7fd-005056ae23aa?accesskey=35c17994cd78000521aeaa3078a63d7441dc365265d1c9e1008b61e41577cea2. Source data for graphs and immunoblots are provided with this paper as a Source Data file. Original microscopy images used to compose figures and for analysis are available through Zenodo under https://doi.org/10.5281/zenodo.16618585. Source data are provided with this paper.

## Code availability
The code is available on Zenodo, a publicly accessible web archive for data and code. The DOI is https://doi.org/10.5281/zenodo.15708552.

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

## Acknowledgements

All the LC-MS/MS experiments including data analysis were performed at the Purdue Proteomics Facility, Bindley Bioscience Center, Purdue University. The authors would also like to thank Jacquelyn Turcinovic at Boston University for assistance with R script writing. Funding support was provided by NIH/NIAID grants R01AI114814 to D.J.L. and R.A.D. and P01AI120943 to D.J.L., C.F.B., D.W.L. and R.A.D.

## Author contributions

C.J.D., A.K., N.T., L.W., and S.H.S. designed and performed experiments, analyzed data, and prepared figures. S.B.S. and C.G.W. contributed to experimental design, data acquisition, and figure preparation. C.D.K. and U.K.A. performed proteomic analyses. C.J.D. wrote the manuscript and developed computational workflows with input from R.A.D. D.W.L. and C.F.B. provided critical reagents, supervision of experimental design, and wrote the manuscript. D.J.L. and R.A.D. conceived the study, supervised all aspects of the work, and wrote the manuscript with input from all authors. All authors reviewed and approved the final manuscript.

## Competing interests

The authors declare no competing interests.
