## [Transparent Peer Review file · Nature Communications]

A protein-proximity screen reveals Ebola virus co-opts the mRNA decapping complex through the scaffold protein EDC4

Corresponding Author: Professor Robert Davey

Version 0:

Reviewer comments:

Reviewer #1

(Remarks to the Author)

This work by Donahue and colleagues presents a comprehensive study of the interactome of six EBOV proteins (NP, VP35, VP40, VP30, VP24, and L) in human cells. For this, the authors used two different strategies (Flag and Strep tag-based AP-MS and TurboID proximity labeling methods) to tag each of the EBOV proteins and determined their interactomes in HRK293T transfected cells using mass spectrometry technology to analyze the proteins present in the corresponding pull-down samples from transfected cells.

The authors have performed well-designed and solid experiments to generate the interactomes for each of the EBOV proteins and have used appropriate bioinformatic tools to extract biological information from the interactome results. There are several published papers describing the interactomes of EBOV proteins, but the present work by Donahue and colleagues has incorporated the unique component of mapping the host protein hits identified in the BioID screening onto the Human Integrated Protein-Protein Interaction Reference and the Prize Collecting Steiner Forest network algorithm to generate a high stringency network to reveal connections between EBOV-host PPI and cellular pathways, and their potential biological implications. Results from these bioinformatic analysis led the authors to focus on the characterization of the interaction of VP35 with the host mRNA decapping complex.

The experimental section of the paper has been superbly executed and the manuscript is clearly written. The authors should be commended for the extensive and high quality of work done to generate a very valuable dataset for the EBOV field. In addition, the authors have presented some interesting findings supporting the potential role the host mRNA decapping complex in modulating levels of EBOV gene expression and replication.

The authors have provided compelling biochemical (CoIP) and imaging (sub-cellular co-localization) evidence that VP35 recruits components of the host mRNA decapping complex via interacting with EDC4 that serves as a scaffold component that binds DCP1 and DCP2. Moreover, the authors have presented evidence that RNAi-mediated knock-down of EDC4 results in reduced levels of EBOV both transcription and replication. However, it was unclear to me whether these findings reflect a specific role of host mRNA decapping machinery in EBOV infection, or that EBOV mRNAs are subjected to the same regulatory process that applies to host cellular mRNAs. The authors should consider the possibility that reduced levels of NP and GP mRNA in cells subjected to RNAi-mediated knock-down of EDC4, reflect enhanced targeting activity of the regulator of mRNA stability MARF1, which activity is negatively regulated by EDC4.

A general limitation of this very elegant work is that the interactomes have been generated based on over-expression of individual EBOV proteins, and some of the identified interactions may not reflect the conditions operating in EBOV-infected cells. It would be very informative to assess the effect on EDC4 knock-down on EBOV multi-step growth kinetic where production of infectious progeny is assessed over time.

Despite the superb technical quality of the present work, there are some technical issues that need some additional discussion:

1) It would be important to document that for EBOV proteins for which functional assays are established, the tagging strategies used by the authors did not result in impaired functionality or the subcellular location of the protein that is seen in EBOV-infected cells. This is relevant because of the size of a TurboID tag ~35 kDa is similar to the size of VP35 and bigger

than VP24 and VP30, which results in a bulky tag that could influence the behavior of a small viral protein.

2) It is important to consider that a requirement of the TurboID is the necessity of K in the binding surface of both proteins and not all proteins that bind to the EBOV proteins will necessarily have K-rich regions. This potential limitation should be discussed.

(Remarks on code availability)

Reviewer #2

(Remarks to the Author)

This is a study by Robert Davey and Douglas LaCount and their co-workers that investigates interacting partners of Ebola virus (EBOV) proteins using proximity labeling with BioID biotin ligase. The authors created stable cell lines expressing each EBOV protein (except the glycoprotein) that were tagged on either N-or C-terminus, and the biotinylated neighboring proteins were identified using LC-MS/MS analysis. The authors used SAINT express workflow followed by Prize-Collecting Steiner Forest algorithm that allowed building a focused network of interacting proteins. Compared to STRING that showed no clear networks, this allowed for identification of novel interacting partners for EBOV proteins. The authors focused on the VP35-interacting DCP and EDC proteins to validate their findings. EDC4 is scaffold that assembles mRNA decapping complex that has been previously linked to viral infection but not to the EBOV infection. The authors showed that VP35 interacts with EDC4 through the C-terminal domain of VP35 and that depletion of the members of this complex prevented synthesis of EBOV mRNA and also anti-genome and genome RNA and blocked viral replication. IN EBOV infected cells, VP35 colocalized with EDC4 and this localization changes during the course of viral infection. Overall, this is an interesting study that moves forward the EBOV field. The study provides an interesting pathway for future proteomics proximity studies using SAINT workflow and Prize-Collecting Steiner Forest algorithm. The reviewer has only some minimal critics outlined below.

Major comments

1. Please describe in more detail the SAINT and Prize-Collecting Steiner Forest algorithms, so that it will be clearer to the general readers.
2. How does your approach compare to a more advanced pathway analysis, such as IPA for example?
3. Lane 464, please describe LC-MS running details.

Minor comments

Lane 464, "LC-MS/MS data was analyzed"... change to "LC-MS/MS data were analyzed"

(Remarks on code availability)

R script is deposited and available

Reviewer #3

(Remarks to the Author)

Review:

Donahue, Kesari et al. A protein-proximity screen reveals Ebola virus co-opts the mRNA decapping complex through the scaffold protein EDC4.

Donahue, Kesari and colleagues performed BioID-based proteomics to analyze the cellular interactome of five Ebola virus protein (NP, VP35, VP30, VP24, VP40). They utilized a network algorithm, which takes into account known human protein-protein interactions from published work, to expand their experimentally determined host-virus interaction network. In identified interactomes, they chose to validate one hit, EDC4, found to interact with Ebola VP35. They revealed that a EDC4-containing, mRNA-decapping complex is important for Ebola virus RNA synthesis. Identification of a mRNA-decapping complex as an Ebola virus host factor is certainly valuable. However, their work did not yet offer mechanistic understanding as to why Ebola virus needs an mRNA-decapping complex for viral RNA synthesis. The following suggestions are intended to improve the quality of this study.

Major comments:

1. The current manuscript has a similar experimental setup (Ebola virus protein interactomes) with a previous interactome study (Batra et al. 2018, PMID: 30550789). One uses proximity labeling and the other used affinity tag-purification. The author should discuss what has been improved in the current work (such as coverage, robustness, fidelity, relevance) compared to Batra et al. 2018. A direct comparison between their findings is missing in the current manuscript.
2. Proximity biotinylation is biased by tagging strategy.
 - As the author pointed out, N- vs C- term tagging of the same viral protein results in different proximity biotinylation profiles (Figure 1c and Figure S1), meaning there could be some false positive hits. It makes more sense to show/highlight those 52 interactors (identified by both N- and C-term tagging of the same viral bait experimentally determined by this study) in Figure 1d. Given the abovementioned technical bias from BioID-tagging, the author should validate a selection of hits per viral bait with untagged protein.
3. Graphical presentation for interactome network is obscure.

- I did not follow the process expanding from 52 experimentally determined hits (line 78) to 447 interactions (line 108) shown in Figure 1d.
 - Figure 2a is not intuitive. If nodes 1 and 4 are hits, what new information does it bring by connecting through blue vs. red path?
 - Consider using different colors or groups to reflect where nodes come from (experimentally determined in this study vs. secondary interactions based on literature vs. from other proteomic works).
 - Same idea with Figure 2c and 2d. It seems to me that a new way of data analysis (PCSF) leveraged known protein-protein interactions in published literature to expand the scope of experimentally determined proximity interactomes and assign weight (a numerical parameter) to each interaction. Could the authors explain what an edge shown in Figure 2c and 2d means, what the weight for each edge is, and whether it is inferred from prior literature (association) or from experiments in this paper (interaction).
 - Needs a legend to show what the different node size means in Figure 2c and 2d.
4. Immunofluorescence microscopy needs quantification.
- Figure 3 c and 3d: not all EDC4 puncta associate with VP35 puncta. Should quantify the level of association.
 - Figure 3e: uninfected cells have background. Should quantify and show statistical difference in PLA signal between infected cells and mock.
5. Lack of growth kinetics data to evaluate the function importance of decapping complex for Ebola infection.
- Current functional experiments are based on the level of viral mRNA synthesis, it is unclear whether mRNA decapping complex can impact the viral growth kinetic (infectious particle production).

Specific comments:

1. Line 9, line 318: L did not express; GP was not included in the proteomic experiment. I would say the study only screened PPI networks for 5 viral proteins (NP, VP35, VP30, VP24, VP40), not 6. Same problem with the statement "screen using all EBOV proteins except GP" in discussion.
2. Line 29: RdRP complex usually refer to L and VP35, which by definition show RNA-dependent RNA polymerase activity in vitro (without NP and VP30). vRNP complex: RdRP plus NP and VP30.
3. Figure 3b shows individually cropped western blots. Need to show uncropped blots.
4. Line 190: specify which cell line was used for infection?
5. Line 219, 267: "vRNA was then visualized by RNAFISH", but Figure 4b shows mRNA. Same with "vRNAFISH". vRNA usually means the viral genomic RNA for negative-strand RNA virus.
6. Figure 5b: should mark molecular weight in western blots
7. Scale bar unit in multiple figure legend should be μm (micrometer) instead of M (micromolar).

(Remarks on code availability)

n/a

Version 1:

Reviewer comments:

Reviewer #1

(Remarks to the Author)

In this revised version of a paper by Donahue and colleagues, the authors have adequately addressed all the issues I raised during the review of the originally submitted paper, and have incorporated the appropriate changes in the revised version of their paper.

I do not have any additional comments regarding the scientific content of this paper.

(Remarks on code availability)

I do not have the level of expertise in coding to assess this component of the paper.

Reviewer #2

(Remarks to the Author)

All comments are answered

(Remarks on code availability)

I could not find the data on the zenodo

Reviewer #3

(Remarks to the Author)

The authors addressed only some of my concerns.

Extended figure 1: In panel a, I consider VP30-immunofluorescence signal here marks viral inclusions formed by the wild-type EBOV. BioID-tagged NP-NT and NP-CT locate completely outside of the VP30-positive subcellular areas, meaning tagged NP cannot participate in a functional viral infection. In panel c, some constructs (NP-NT, VP30-NT, VP35-CT) only have 10-20% of the activity of untagged proteins. These data undermine the relevance of resulting interactomes for NP, VP35 and VP30, given the tagged proteins could be mislocated or dysfunctional. Data interpretation in line 68-77 is inaccurate and incomplete.

Typically, colocalization of two channels is quantified by Pearson's correlation coefficient or Mander's colocalization coefficient. Not the kind of quantification shown in Figure 3d. Even if the signal EDC4 is randomly distributed in the cells, as you increase any random subcellular region of interest (ROI), one would expect more EDC4 dots fall into a larger ROI. Figure 3d doesn't support VP35-EDC4 interactions in cells.

For clarity, show single channel image for Figure 3c, 6a, and 6b. Label each color in the merged image in extended figure table 4.

Figure 4: growth curve should show the full log scale. Right now, the Y-axis is at $10E5$. Projecting the current result back to log scale, the difference between control and knock-down is less than one log. Single knock-down of EDC4 has modest effect on viral growth. Given the data shown in Figure 6, perhaps depleting multiple components in the decapping complex will have a greater impact on viral growth?

All bar graphs should show error bars on both sides, also should show individual data points instead of one bar.

(Remarks on code availability)

Version 2:

Reviewer comments:

Reviewer #3

(Remarks to the Author)

My concerns have been addressed.

(Remarks on code availability)

REVIEWER COMMENTS

Response to review.

Below is a detailed summary of the work we performed since receiving reviews and other responses to the reviewer comments. We thank the reviewers for taking the time to review our manuscript and for their helpful comments. We have addressed each of the comments and believe the manuscript is much improved. Through responding to these comments, the manuscript was edited to improve clarity. Below we have included all of the reviewers' comments and our responses indicated by ">" symbol.

Reviewer #1 (Remarks to the Author):

This work by Donahue and colleagues presents a comprehensive study of the interactome of six EBOV proteins (NP, VP35, VP40, VP30, VP24, and L) in human cells. For this, the authors used two different strategies (Flag and Strep tag-based AP-MS and TurboID proximity labeling methods) to tag each of the EBOV proteins and determined their interactomes in HRK293T transfected cells using mass spectrometry technology to analyze the proteins present in the corresponding pull-down samples from transfected cells.

The authors have performed well-designed and solid experiments to generate the interactomes for each of the EBOV proteins and have used appropriate bioinformatic tools to extract biological information from the interactome results. There are several published papers describing the interactomes of EBOV proteins, but the present work by Donahue and colleagues has incorporated the unique component of mapping the host protein hits identified in the BioID screening onto the Human Integrated Protein-Protein Interaction Reference and the Prize Collecting Steiner Forest network algorithm to generate a high stringency network to reveal connections between EBOV-host PPI and cellular pathways, and their potential biological implications. Results from these bioinformatic analysis led the authors to focus on the characterization of the interaction of VP35 with the host mRNA decapping complex.

The experimental section of the paper has been superbly executed and the manuscript is clearly written. The authors should be commended for the extensive and high quality of work done to generate a very valuable dataset for the EBOV field. In addition, the authors have presented some interesting findings supporting the potential role the host mRNA decapping complex in modulating levels of EBOV gene expression and replication.

The authors have provided compelling biochemical (CoIP) and imaging (sub-cellular co-localization) evidence that VP35 recruits components of the host mRNA decapping complex via interacting with ECD4 that serves as a scaffold component that binds DCP1 and DCP2. Moreover, the authors have presented evidence that RNAi-mediated knock-down of EDC4 results in reduced levels of EBIOV both transcription and replication. However, it was unclear to me whether these findings reflect a specific role of host mRNA decapping machinery in EBOV infection, or that EBOV mRNAs are subjected to the same regulatory process that applies to host cellular mRNAs.

- We thank the reviewer for their supportive comments. We agree that it will be interesting to further explore how the decapping complex is altering EBOV RNA levels. What we have shown in this manuscript is that depletion of decapping complex members results in

decreased viral RNA levels, suggesting that viral RNAs are regulated in a manner that is distinct from cellular RNAs. We have made sure to address this in the discussion, lines 494-501.

The authors should consider the possibility that reduced levels of NP and GP mRNA in cells subjected to RNAi-mediated knock-down of EDC4, reflect enhanced targeting activity of the regulator of mRNA stability MARF1, which activity is negatively regulated by EDC4.

- Through addressing this comment, it was realized that the gene name database that was being used was outdated and so, multiple genes, like MARF1 were not represented. This has now been corrected. Reanalysis of the screening data revealed that MARF1 was identified as a hit, and network mapping positioned it immediately adjacent and connected to the decapping complex subnetwork in a subnetwork related to endocytic trafficking. To reflect this, we have updated the figures and the discussion (Figure 2 and lines 494-501 of the discussion), describing potential roles of MARF1 as well as XRN1 as accessory factors that control RNA stability

A general limitation of this very elegant work is that the interactomes have been generated based on over-expression of individual EBOV proteins, and some of the identified interactions may not reflect the conditions operating in EBOV-infected cells. It would be very informative to assess the effect on EDC4 knock-down on EBOV multi-step growth kinetic where production of infectious progeny is assessed over time.

- As requested, we performed a multi-step growth curve in the EDC4 KD cells. This is now shown in Figure 4e and described in the results (lines 276-280). We find that EDC4 depletion results in a significant decrease in virus production over a multiple cycle infection.

Despite the superb technical quality of the present work, there are some technical issues that need some additional discussion:

1) It would be important to document that for EBOV proteins for which functional assays are established, the tagging strategies used by the authors did not result in impaired functionality or the subcellular location of the protein that is seen in EBOV-infected cells. This is relevant because of the size of a TurboID tag ~35 kDa is similar to the size of VP35 and bigger than VP24 and VP30, which results in a bulky tag that could influence the behavior of a small viral protein.

- We performed 2 new analyses of the tagged proteins that evaluated their behavior to associate with other virus proteins, testing ability to form virus inclusion bodies at sites of virus replication and functionality in the virus RNA dependent RNA polymerase ribonuclear complex (rVNP) for which most of the proteins participate. Proteins were expressed in trans in infected cells and then colocalization in inclusion bodies and other virus induced structures were observed (extended data Figure 1a-b and lines 65-76). The second test evaluated performance of each tagged protein in supporting replication of a virus minigenome. The minigenome system (TrVLP) allowed substitution of each component of the rVNP from plasmids (extended data figure 1c, lines 76-80). Both experiments were useful and highlighted that in general, the tags were permissive for functions of each

protein. This information adds greatly to the manuscript and will be useful for the reader in better understanding how tagging influenced host factor interaction.

2) It is important to consider that a requirement of the TurboID is the necessity of K in the binding surface of both proteins and not all proteins that bind to the EBOV proteins will necessarily have K-rich regions. This potential limitation should be discussed.

- We agree that lysine availability of target proteins is a limitation using BioID. This has now been discussed in lines 387-391.

Reviewer #2 (Remarks to the Author):

This is a study by Robert Davey and Douglas LaCount and their co-workers that investigates interacting partners of Ebola virus (EBOV) proteins using proximity labeling with BioID biotin ligase. The authors created stable cell lines expressing each EBOV protein (except the glycoprotein) that were tagged on either N-or C-terminus, and the biotinylated neighboring proteins were identified using LC-MS/MS analysis. The authors used SAINT express workflow followed by Prize-Collecting Steiner Forest algorithm that allowed building a focused network of interacting proteins. Compared to STRING that showed no clear networks, this allowed for identification of novel interacting partners for EBOV proteins. The authors focused on the VP35-interacting DCP and EDC proteins to validate their findings. EDC4 is scaffold that assembles mRNA decapping complex that has been previously linked to viral infection but not to the EBOV infection. The authors showed that VP35 interacts with EDC4 through the C-terminal domain of VP35 and that depletion of the members of this complex prevented synthesis of EBOV mRNA and also anti-genome and genome RNA and blocked viral replication. IN EBOV infected cells, VP35 colocalized with EDC4 and this localization changes during the course of viral infection. Overall, this is an interesting study that moves forward the EBOV field. The study provides an interesting pathway for future proteomics proximity studies using SAINT workflow and Prize-Collecting Steiner Forest algorithm. The reviewer has only some minimal critics outlined below.

Major comments

1. Please describe in more detail the SAINT and Prize-Collecting Steiner Forest algorithms, so that it will be clearer to the general readers.

- To add more clarity about SAINT scoring and the algorithms used, we have added additional information about the SAINT algorithm on lines 91-93 and Prize-Collect Steiner Forest (PCSF) algorithm on lines 149-160. Additionally, we have clarified the language in the legend of figure 2a regarding PCSF and have added additional keys to Figure 2.

2. How does your approach compare to a more advanced pathway analysis, such as IPA for example?

- We have more clearly addressed the differences between our analysis and other commonly performed network analyses (lines 139-148). IPA is best suited to gene expression data and uses a proprietary algorithm and so is difficult to compare to. Instead STRING and Metascape are commonly used for PPI analysis and are now compared to PCSF. We have performed comparative analyses with STRING (Supplemental figure 1) and Metascape (supplemental figure 2). Metascape uses a traditional edge-density algorithm MCODE to identify protein complexes. Metascape identified proteins related to the decapping complex but our

approach improved detection of more complete complexes and connected complexes to other related proteins as subnetworks. This comparison is described on lines 139-148 and discussed on lines 459-479.

3. Lane 464, please describe LC-MS running details.

- We have extended the information about the LC-MS running details, and those can now be found on lines 539-557.

Minor comments

Lane 464, "LC-MS/MS data was analyzed"... change to "LC-MS/MS data were analyzed"

- This correction has been made.

Reviewer #2 (Remarks on code availability): R script is deposited and available

Reviewer #3 (Remarks to the Author):

Review: Donahue, Kesari et al. A protein-proximity screen reveals Ebola virus co-opts the mRNA decapping complex through the scaffold protein EDC4.

Donahue, Kesari and colleagues performed BioID-based proteomics to analyze the cellular interactome of five Ebola virus protein (NP, VP35, VP30, VP24, VP40). They utilized a network algorithm, which takes into account known human protein-protein interactions from published work, to expand their experimentally determined host-virus interaction network. In identified interactomes, they chose to validate one hit, EDC4, found to interact with Ebola VP35. They revealed that a EDC4-containing, mRNA-decapping complex is important for Ebola virus RNA synthesis. Identification of a mRNA-decapping complex as an Ebola virus host factor is certainly valuable. However, their work did not yet offer mechanistic understanding as to why Ebola virus needs an mRNA-decapping complex for viral RNA synthesis. The following suggestions are intended to improve the quality of this study.

Major comments:

1. *The current manuscript has a similar experimental setup (Ebola virus protein interactomes) with a previous interactome study (Batra et al. 2018, PMID: 30550789). One uses proximity labeling and the other used affinity tag-purification. The authors should discuss what has been improved in the current work (such as coverage, robustness, fidelity, relevance) compared to Batra et al. 2018. A direct comparison between their findings is missing in the current manuscript.*

- Batra et al. used immunoprecipitation to pull down host proteins directly binding to virus proteins. In general, only the tightest interactions are preserved and proteins that are not directly bound are often not captured. BioID labels up such secondary proteins in complexes. We have added this difference as part of a greater discussion to other screens on lines 394-408 and provide a graphical comparison of all such interaction analyses in Supplemental figure 5.

2. *Proximity biotinylation is biased by tagging strategy.*

- *As the author pointed out, N- vs C- term tagging of the same viral protein results in different proximity biotinylation profiles (Figure 1c and Figure S1), meaning there could be some false positive hits. It makes*

more sense to show/highlight those 52 interactors (identified by both N- and C-term tagging of the same viral bait experimentally determined by this study) in Figure 1d. Given the abovementioned technical bias from BioID-tagging, the author should validate a selection of hits per viral bait with untagged protein.

- We thank the reviewer for the suggestion and have added in several pieces of information to address this concern. While there is some overlap in N- and C-terminally tagged constructs from our screening, we find that by including both N-terminal and C-terminal identified interactors in our network we are able to identify whole complexes that would otherwise be missing. While it is likely that N-terminal and C-terminal BioID tagging disrupt identification of host protein interactions dependent on the respective protein termini, the tag on the opposite end should make up for this loss. Most relevantly for this manuscript, the decapping complex members were identified only with N-terminally tagged VP35, and therefore emphasis on only the overlapping hits would miss this biologically relevant complex. We have included this observation in lines 65-88.
- To evaluate the impact of each tag on virus protein function, we have now included cellular localization and trVLP functionality of both N- and C-terminally tagged BioID tagged viral proteins to support the notion that tagging proteins does not render them unfunctional. We have discussed this point in the response to Reviewer #1's first main point. This work is added in lines 65-88 of the results.
- Our network analysis identified multiple proteins and complexes that are well characterized interactors and modulators of EBOV infection, validating our approach. We have expanded this discussion on lines 180-189 to give examples for multiple viral proteins.

3. Graphical presentation for interactome network is obscure.

• I did not follow the process expanding from 52 experimentally determined hits (line 78) to 447 interactions (line 108) shown in Figure 1d.

- We agree the description was not clear. We have clarified what was done for the experimental design and analysis on lines 102-105.

4. Figure 2a is not intuitive. If nodes 1 and 4 are hits, what new information does it bring by connecting through blue vs. red path?

- The blue route, while longer, is more well supported by the literature, incorporates more hits found in the screen, and incorporates only one node not found in the screen (a Steiner node), and therefore this pathway is preferred because it has a higher confidence. Conversely, while the red path is shorter and has less connections between the hits, the pathway to connect them is less supported by the literature and requires incorporation of multiple nodes that were not found in the hit list. To clarify we have extended the explanation of this on lines 154-160 and in the Figure 2 legend.

• Consider using different colors or groups to reflect where nodes come from (experimentally determined in this study vs. secondary interactions based on literature vs. from other proteomic works).

- We have added a sentence to the figure legend for Figure 2 as well adding a legend in the figure to explain the relevance of node sizes.

- Same idea with Figure 2c and 2d. It seems to me that a new way of data analysis (PCSF) leveraged known protein-protein interactions in published literature to expand the scope of experimentally determined proximity interactomes and assign weight (a numerical parameter) to each interaction. Could the authors explain what an edge shown in Figure 2c and 2d means, what the weight for each edge is, and whether it is inferred from prior literature (association) or from experiments in this paper (interaction).

- We have edited the description of what the edges and nodes are derived from within our network analysis. We have also added a legend in Figure 2c that shows how node size corresponds to the SAINT score, a measure of hit strength. This is further described in figure 2 legend and edge metrics in lines 154-160 and 170-171. We have additionally listed edge weights for network interactions in Extended data table 3

- *Needs a legend to show what the different node size means in Figure 2c and 2d.*

- This has been added in bottom left of the figure 2.

1. Immunofluorescence microscopy needs quantification.

- Figure 3 c and 3d: not all EDC4 puncta associate with VP35 puncta. Should quantify the level of association.

- Based on this comment, we decided to perform super-resolution confocal microscopy as this technique is well matched to the resolution of the EDC4 and virus complexes that we were seeing and would enable stronger quantitation than from the original microscopy images. We are excited to show this new data in Figures 3, and 6 and provide one of the TIFF z-stacks in Extended data figure 4. The super-resolution microscope, having additional sensitivity allowed us to resolve discrete complexes of EDC4 and VP35 in the cell cytoplasm and EDC4 within inclusion bodies. Quantitation, as requested, revealed a stoichiometric relationship of inclusion body volume to the number of discrete EDC4, DCP1A and DCP2 puncta, indicating some higher order structure within the inclusion bodies. These findings are described on lines 227-229, and figure 3d and Supplemental figure 4.

- Figure 3e: uninfected cells have background. Should quantify and show statistical difference in PLA signal between infected cells and mock.

- As requested we have quantified the PLA signal and demonstrated a significant difference between EBOV infected and mock-infected samples in Figure 3e-f, and in Extended data figure 4c and described on lines 230-232.

2. *Lack of growth kinetics data to evaluate the function importance of decapping complex for Ebola infection.*

- *Current functional experiments are based on the level of viral mRNA synthesis, it is unclear whether mRNA decapping complex can impact the viral growth kinetic (infectious particle production).*

- Reviewer #1 also requested this experiment. We performed a multi-step growth curve in the ECD4 KD cells. This is now shown in Figure 4e and described in the results (lines 276-280). We find that EDC4 depletion results in a significant decrease in virus production over a multiple cycle infection.

Specific comments:

1. Line 9, line 318: L did not express; GP was not included in the proteomic experiment. I would say the study only screened PPI networks for 5 viral proteins (NP, VP35, VP30, VP24, VP40), not 6. Same problem with the statement “screen using all EBOV proteins except GP” in discussion.

- The N-terminally tagged L was expressed, albeit weakly, but this is typical for this protein. Importantly, it did specifically tag one protein and so, we feel it important to include this observation. With this observation, we have evaluated 6 of the seven virus proteins. We now comment on the expression levels of L and each other protein on lines 61-65.

2. Line 29: RdRP complex usually refer to L and VP35, which by definition show RNA-dependent RNA polymerase activity in vitro (without NP and VP30). vRNP complex: RdRP plus NP and VP30.

- We thank the reviewer for this correction and have addressed this comment in lines 29-30.

3. Figure 3b shows individually cropped western blots. Need to show uncropped blots.

- Uncropped western blots are now provided in Extended data figure 2 (biotinylated protein blots) and supplemental Figure 6 (for figure 3a,b).

Line 190: specify which cell line was used for infection?

- We used HeLa cells for the indicated infection, and this information has been added to the manuscript at line 316 and in the methods.

5. Line 219, 267: “vRNA was then visualized by RNAFISH”, but Figure 4b shows mRNA. Same with “vRNAFISH”. vRNA usually means the viral genomic RNA for negative-strand RNA virus.

- These corrections have been made In figure 4b and elsewhere.

3. Figure 5b: should mark molecular weight in western blots

- Molecular weights have been added to the blots in Figure 5b

Scale bar unit in multiple figure legend should be mm (micrometer) instead of mM(micromolar).

- The scale bar units have been adjusted to micrometers (um). We thank the reviewer for this correction.

4. *Reviewer #3 (Remarks on code availability): n/a*

REVIEWER COMMENTS

Reviewer #1 (Remarks to the Author):

I do not have any additional comments regarding the scientific content of this paper.

Reviewer #2 (Remarks to the Author):

All comments are answered

Reviewer #3 (Remarks to the Author):

Extended figure 1: In panel a, I consider VP30-immunofluorescence signal here marks viral inclusions formed by the wild-type EBOV. BioID-tagged NP-NT and NP-CT locate completely outside of the VP30-positive subcellular areas, meaning tagged NP cannot participate in a functional viral infection.

Response: Concern was raised whether some BioID2-tagged constructs (e.g., NP-NT, VP30-NT, VP35-CT) might be mislocated or dysfunctional, as they were observed outside of VP30-positive viral inclusions in Extended Figure 1a (and what is also now shown in Figure 1). To clarify, our BioID analyses were conducted in uninfected cells expressing individual tagged Ebola virus proteins, which we have further emphasized in the text. However, to provide additional validation, we examined the localization of these tagged proteins during authentic EBOV infection to show that each was capable of interacting with other virus proteins in situ. While NP-expressing cells showed fewer VP30-positive co-inclusions than for the other tagged constructs, colocalization was evident in cells expressing lower levels of NP fusion proteins. Additionally, the NP constructs alone formed inclusion-like structures consistent with NP localization seen after authentic virus infection and previously reported by ourselves and others. The reduced infection rates in BioID2-tagged NP-expressing cells likely result from prior NP expression interfering with normal infection rather than loss of function and is supported by work with recombinant virus systems where the ratio of NP and other virus proteins needs to be carefully controlled to obtain infectious virus. Importantly, NP-NT and NP-CT both exhibited significant activity in minigenome assays, demonstrating their ability to support viral RNA replication, which is of strong interest to the virology readership. We have added these caveats to the manuscript text.

In response to both the requests, we further validated the inclusion of these constructs in our dataset by comparing our interactions with the Carlson et al. (2024) functional screen. As stated above, 31% of the proteins identified in our BioID dataset were also hits in at least one Carlson et al. functional assay, exceeding the 21% baseline across

their dataset. This enrichment provides additional support for the biological relevance of the identified interactome, including interactions identified by BioID2-tagged NP. The comparison to this and other previous interaction analyses are now shown in Extended data figure 1c and summarized in Figure 7.

In panel c, some constructs (NP-NT, VP30-NT, VP35-CT) only have 10-20% of the activity of untagged proteins. These data undermine the relevance of resulting interactomes for NP, VP35 and VP30, given the tagged proteins could be mislocated or dysfunctional. Data interpretation in line 68-77 is inaccurate and incomplete.

Response: We have now removed interactions from the network analytics for those proteins with reduced activity, low expression or aberrant localization (NT-L, NT-VP30, CT-VP35, and CT-VP40). For completeness, and because some of these excluded constructs identified previous characterized interactions (see below), all the data on interactions is retained in Extended data table 1, which includes additional columns showing activity in the mini genome assay and whether the interaction was included in the network analyses.

For reference, NT-NP (25% of wild-type activity), CT-NP (66%), CT-VP30 (103%), and NT-VP35 (64%) exhibited significant activity ($p = 0.01$ for NT-NP, $p < 0.0001$ for the others; one-way ANOVA), demonstrating their ability to support replication. While NT-VP30 (18%) and CT-VP35 (9%) had lower activity, both consistently yielded values above the negative control, albeit without statistical significance. To improve transparency, we added asterisks to Figure 1C to indicate statistically significant differences in minigenome activity and revised the text to clarify these observations.

N-terminal fusions to VP30 are known to reduce VP30 function. Similarly, C-terminal fusions to VP35 are likely to disturb VP35 folding since the C terminus is buried in the VP35 crystal structure. Both constructs were expressed at lower levels than their CT- and NT-tagged counterparts, respectively, which may contribute to their lower activity in the minigenome assay.

We note that the constructs with reduced activity identified previously validated interactions as well as less characterized interactions reported in other large-scale studies. For example, the N-terminal fusion to VP30 was the only construct that identified SRPK2 as a binding partner of VP30. SRPK2 binds to VP30 and phosphorylates serine residues in the VP30 N terminal domain. (SRPK1 was also identified by NT-VP30 but fell below our stringent cutoffs and was therefore excluded from the final dataset.) Similarly, NT-NP labelled PPP2R2A, which is a regulatory subunit of the B56 phosphatase that binds to NP and dephosphorylates VP30.

We added text and Extended data figure 1c to better illustrate these points.

Typically, colocalization of two channels is quantified by Pearson's correlation coefficient or Mander's colocalization coefficient. Not the kind of quantification shown in Figure 3d. Even if the signal EDC4 is randomly distributed in the cells, as you increase

any random subcellular region of interest (ROI), one would expect more EDC4 dots fall into a larger ROI. Figure 3d doesn't support VP35-EDC4 interactions in cells.

Response: We thank the reviewer for the input on this observation. We agree, especially now after using super-resolution confocal microscopy that the EDC4 puncta are seemingly randomly distributed in the cell cytoplasm with no clear association with VP35 puncta suggesting interaction may be through population of the proteins that is not readily identified by microscopy. Nevertheless, these puncta are also present within inclusion bodies and are not excluded. Furthermore, Proximity Ligation Assays indicate that the proteins are in close proximity within the cell and the immunoprecipitation work indicates the ability to associate in a complex that is strong enough to survive extraction from cells. We have revised the manuscript to acknowledge each of these outcomes.

For clarity, show single channel image for Figure 3c, 6a, and 6b. Label each color in the merged image in extended figure table 4.

Response: This is done. We have also provided optical slices from confocal images that show EDC4 puncta inside inclusion bodies.

Figure 4: growth curve should show the full log scale.

Response: This is done.

Single knock-down of EDC4 has modest effect on viral growth. Given the data shown in Figure 6, perhaps depleting multiple components in the decapping complex will have a greater impact on viral growth?

Response: We agree that depleting multiple components of the decapping complex may have a greater impact on viral replication, and plan to pursue those experiments in our next study. We have expanded the discussion to acknowledge this as a future research direction.

All bar graphs should show error bars on both sides, also should show individual data points instead of one bar.

Response: This is done. In general, all figures have been edited for consistency.